# Abnormal strong burn-in degradation of highly efficient polymer solar cells caused by spinodal donor-acceptor demixing

Ning Li[1], José Darío Perea[1], Thaer Kassar[2], Moses Richter[1], Thomas Heumueller[1], Gebhard J. Matt[1], Yi Hou[1,3], Nusret S. Güldal[1], Haiwei Chen[1], Shi Chen[1], Stefan Langner[1], Marvin Berlinghof[2], Tobias Unruh[2] & Christoph J. Brabec[1,3,4]

The performance of organic solar cells is determined by the delicate, meticulously optimized bulk-heterojunction microstructure, which consists of finely mixed and relatively separated donor/acceptor regions. Here we demonstrate an abnormal strong burn-in degradation in highly efficient polymer solar cells caused by spinodal demixing of the donor and acceptor phases, which dramatically reduces charge generation and can be attributed to the inherently low miscibility of both materials. Even though the microstructure can be kinetically tuned for achieving high-performance, the inherently low miscibility of donor and acceptor leads to spontaneous phase separation in the solid state, even at room temperature and in the dark. A theoretical calculation of the molecular parameters and construction of the spinodal phase diagrams highlight molecular incompatibilities between the donor and acceptor as a dominant mechanism for burn-in degradation, which is to date the major short-time loss reducing the performance and stability of organic solar cells.

[1] Institute of Materials for Electronics and Energy Technology (i-MEET), Department of Materials Science and Engineering, Friedrich-Alexander University Erlangen-Nürnberg, Martensstrasse 7, 91058 Erlangen, Germany. [2] Chair for Crystallography and Structural Physics, Department of Physics, Friedrich-Alexander University Erlangen-Nürnberg, Staudtstrasse 3, 91058 Erlangen, Germany. [3] Erlangen Graduate School in Advanced Optical Technologies (SAOT), Paul-Gordan-Strasse 6, 91052 Erlangen, Germany. [4] Bavarian Center for Applied Energy Research (ZAE Bayern), Immerwahrstrasse 2, 91058 Erlangen, Germany. Correspondence and requests for materials should be addressed to N.L. (email: Ning.Li@fau.de) or to C.J.B. (email: Christoph.Brabec@fau.de).

Tremendous progress has been made in the field of organic photovoltaics (OPV) in the last few years, and the power conversion efficiencies (PCEs) of organic solar cells (OSCs) were steadily improved to the 11% regime[1–12]. To push the OPV technology towards commercial applications, the reliability and stability of champion OPV devices have to be examined[4,13]. The state-of-the-art OPV devices are based on the bulk-heterojunction (BHJ) structure, where the organic donor and acceptor are fine-mixed in the nanometre regime to facilitate exciton dissociation at the donor/acceptor interface[14]. This regime is frequently referred to as the amorphous regime. Pure donor and acceptor domains are also required to form bi-continuous pathways to sweep out the free charge carriers prior to bulk recombination. These are typically referred to as the crystalline or ordered regimes. The two micro-morphologies, the fine-mixed region and the phase separated region, have to be simultaneously optimized in the delicate BHJ structure to maximize the photovoltaic parameters, such as short circuit current density ($J_{SC}$) and fill factor (FF). The $J_{SC}$ represents the number of photogenerated charge carriers extracted from an OSC, while the FF is influenced by many factors, and generally can be considered as the competition between extraction and recombination of the photo-generated charge carriers. Various processing strategies, such as thermal or solvent treatment, binary or ternary solvent formulation as well as thermally assisted film drying, have been developed for solution-processed OSCs to attain an ideal BHJ microstructure[15–28]. The state-of-the-art OSCs achieved the $J_{SC}$ of over $20\,\mathrm{mA\,cm^{-2}}$ along with the high FFs of over 70%, demonstrating promising ability to effectively harvest the photo-generated charge carriers[21,29–31].

In the last few years, hundreds of new donors with well-designed chemical structures were reported to obtain high efficiencies[5–10,29,32–35]. In 2013, Chen et al.[36] developed a new conjugated polymer, FBT-Th$_4$(1,4), based on 5,6-difluorobenzothiadiazole (FBT) as the acceptor unit and quarterthiophene (Th$_4$) as the donor unit. This polymer showed strong interchain aggregation in solutions at room temperature, and achieved a PCE of 7.6% for optimized OSCs. Later, Liu et al.[21] boosted the PCE to over 10 % by modifying the length of polymer side chains (named as PffBT4T-2OD or PCE11) and optimizing the processing conditions. The attained promising performance can be attributed to the preferential molecular orientation, molecular packing and the sufficient polymer aggregation to yield a favourable micromorphology for polymer donor and fullerene acceptor[23]. Further fine-tuning the polymer side chains and solvent formulation led to a certified record PCE of 11.5% (ref. 29). It is notable to mention that these very high efficiencies achieved for PCE11-based OSCs strongly depend on the film drying conditions, especially the processing temperature. It is very difficult to precisely control the drying temperature of thin films by spin coating, as the substrate is typically not heated during processing. However, the doctor-blading technique, which is compatible with large-scale production, maintains the temperature of substrate and solution, exhibiting great advantages in processing the PCE11-based OSCs[37,38].

Although a very promising PCE was reached by optimizing the BHJ micromorphology, the microstructural stability of the finely mixed and phase-separated regions was not examined. The device lifetime of OSCs is determined by the stability of the favourable but delicate BHJ micromorphology, which is required to be long-term stable under operational conditions[4,39,40]. However, in the case of PCE11 blading, the favourable BHJ micromorphology for highly efficient OSCs was attained by controlling the drying kinetics of each BHJ component, which is generally far away from its thermodynamic equilibrium[41–43]. Therefore, the resulting micromorphology of a BHJ structure is typically metastable, leading to a rapid performance reduction in OSCs when annealed at elevated temperatures[4,13]. To enhance the operational lifetime of OSCs, the meta-stability of the BHJ micromorphology has to be well analysed and understood.

Here we fabricate highly efficient OSCs based on PCE11:[6,6]-phenyl-C$_{61}$-butyric acid methyl ester (PCBM) by doctor-blading and investigate the stability of the optimized micromorphology under various conditions. An abnormal strong burn-in loss, which occurs at room temperature and is independent of storage conditions, is found to be associated with the demixing of the donor and acceptor finely mixed phases. Analysis of the donor–acceptor phase diagram, which is determined from the numerically calculated molecular parameters, confirms the strong tendency for this phase separation.

## Results

**Abnormal strong burn-in degradation.** The chemical structure of the polymer donor, PffBT4T-2OD (PCE11), is depicted in Fig. 1a. Figure 1b illustrates the inverted device architecture, in which solution-processed ZnO nanoparticles and thermally evaporated MoOx are employed as n- and p-type interfacial layers, respectively. All the devices investigated in this work were fabricated by doctor-blading in air under optimized processing conditions; see Methods in the Supporting Information. The $J$–$V$ characteristics of the optimized PCE11:PCBM devices (average over 20 devices) are shown in Fig. 1c. The as fabricated OSCs were evaluated directly after thermal evaporation of the top electrode (abbreviated as fresh). Then the devices were stored in dark under ambient conditions for 5 days and evaluated again under the same conditions (abbreviated as aged). The fresh devices showed an average PCE of 9.2%, which is comparable to the previously reported values and among the highest efficiencies for OSCs printed by roll-to-roll compatible deposition method under ambient conditions[21,44–46]. The photovoltaic parameters extracted from Fig. 1c are summarized in Supplementary Table 1. Surprisingly, an abnormal photocurrent loss by 30–40% was observed for the devices stored in dark within only 5 days, while their $V_{OC}$ and FF remained at the same level. This significant $J_{SC}$ loss in the state-of-the-art OSCs stored in dark at room temperature has not been reported to date. However, it determines whether a BHJ system is qualified for large-scale mass production, and has to be in-depth investigated and analysed.

The small amount additive, 1,8-diiodooctane (DIO), used for tuning the BHJ micromorphology may negatively affect the performance of OSCs[46–48]. To exclude this influence, OSCs without DIO were fabricated and stored in N$_2$ with and without encapsulation. The photovoltaic parameters of the corresponding OSCs are summarized in Supplementary Figs 1 and 2. OSCs processed without additive and without exposure to air also showed a decrease in $J_{SC}$ by $\sim 20\%$, while the FF and $V_{OC}$ values remained again the same. A similar burn-in loss was observed for OSCs based on regular device architecture as well as PC$_{70}$BM as an acceptor. Supplementary Fig. 3 depicts the $J$–$V$ characteristics of OSCs with aged active layer, in which the PCE11:PCBM was aged in dark under ambient conditions for 5 days prior to deposition of the top electrode. The rate of performance loss was found to be independent of the top electrode. Since the abnormal strong $J_{SC}$ burn-in loss was observed in all the aforementioned OSCs, we conclude that the loss indeed results from the degradation of the BHJ microstructure rather than from an interface or an electrode degradation.

Figure 2a shows the evolution of the photovoltaic parameters of a PCE11:PCBM solar cell under continuous 1 sun equivalent

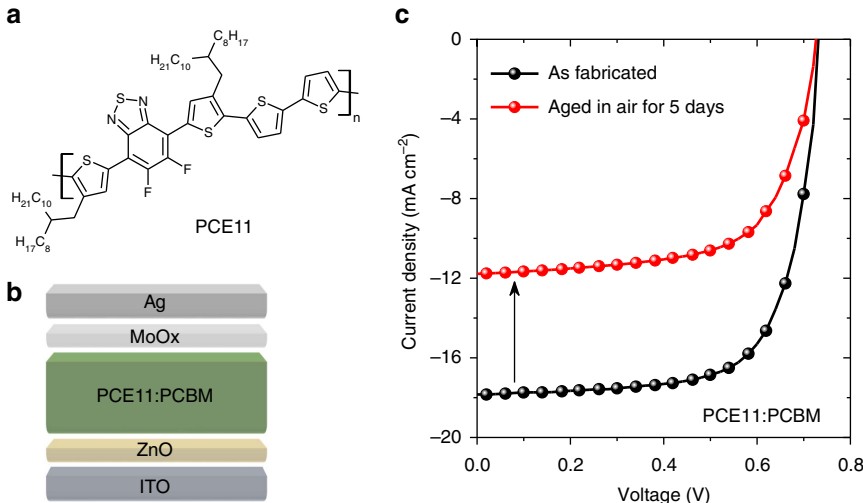

**Figure 1 | Device architecture and J–V characteristics of PCE11:PCBM organic solar cells.** (**a**) The chemical structure of PffBT4T-2OD (PCE11) and (**b**) the inverted device architecture of organic solar cells: ITO/ZnO/PCE11:PCBM/MoOₓ/Ag. (**c**) J–V characteristics of optimized PCE11:PCBM organic solar cells measured after fabrication (fresh) and after aged in air for 5 days (aged).

illumination provided by a white light LED. The OSC was measured in $N_2$ atmosphere and its temperature was determined to be $\sim 330$ K during the measurement. A rapid reduction in $J_{SC}$ was observed within the first 50 h, and then the $J_{SC}$ value remained almost unchanged until the end of the measurement. Over 50 devices were investigated under the continuous illumination, and all showed the similar burn-in losses *in* $J_{SC}$ within the first 50 h. The PCE11:PCBM solar cells fabricated by spin-coating in inert atmosphere were also investigated under the same conditions, and were found to be the same as those fabricated by doctor blading under ambient conditions, as shown in Supplementary Fig. 4. The illumination spectrum of the white LED used for the stability test is shown in Supplementary Fig. 5.

It's notable to mention that the more significant $J_{SC}$ loss was observed for the OSCs stored under ambient conditions, indicating that oxygen and humidity may play an additional role in degradation of OSCs. However, the discussion on photo-oxidation mechanisms in OSCs is beyond the scope of the present work, which is focused on the exploring and explaining the morphology changes in BHJ microstructures for optimized PCE11:PCBM OSCs. The range of performance reduction and the deviation in absolute performance numbers does not affect the discussion and conclusion of the work. Figure 2b summarizes the $J_{SC}$ evolution of OSCs under different storage conditions. The $J_{SC}$ values for 330K were taken from Fig. 2a. The $J_{SC}$ of an OSC stored at room temperature in vacuum ($10^{-6}$ mbar) decreased by $\sim 10\%$ in 72 h, while the $J_{SC}$ of an OSC stored at 220 K, whose BHJ microstructure was frozen at such low temperature, did not show any burn-in loss in $J_{SC}$ during the measurement. The data summarized in Fig. 2b clearly suggest that the abnormal $J_{SC}$ loss in the optimized PCE11:PCBM OSCs is induced by the BHJ micromorphology change, which occurs even at room temperature. We attempted to determine the glass transition temperature ($T_g$) of neat materials and BHJ blends by means of temperature-modulated differential scanning calorimetry (m-DSC). However, as depicted in Supplementary Fig. 6, no clear $T_g$ was observed for neat PCE11 and neat PCBM, while a reversible transition at 14.6 °C was observed for the PCE11:PCBM 1:1 blend from the second heating scan. It is worthwhile to mention that the m-DSC measurement cannot mimic the thermal behaviour of PCE11:PCBM BHJ blend used for OSCs, and the m-DSC samples

prepared from drop-casting without solvent additive may have different morphology compared to that optimized for OSCs.

**Morphology investigation.** EL and FTPS measurements were performed on fresh and aged devices to in-depth explore the changes in the BHJ micromorphology on a molecular scale. EL spectra were measured at a constant applied voltage of 3 V and are shown in Fig. 3a. The fresh sample showed a very weak emission peak at $\sim 1.39$ eV, while the aged sample measured at the same applied voltage showed a broad and intensive emission spectrum in the range of 1.2–1.8 eV. As compared with the singlet emission of PCE11 and PCBM, the EL emission spectrum of the aged sample consists of a broad singlet emission peak of PCE11 at 1.39 eV and further emission contributed by the neat PCBM at 1.7 eV. The EL spectra of fresh and aged PCE11:PCBM samples measured at a constant applied current show the same feature (Supplementary Fig. 7). We also fabricated OSCs incorporating BHJ active layer treated with solvent vapour annealing (SVA) at 100 °C to intentionally create strong phase separation for fresh BHJ films. After deposition of the active layer by doctor blading in air, the wet film was kept at 100 °C for 20 min and covered by a glass lid in the dichlorobenzene atmosphere prior to deposition of the top electrode. Owing to the phase separation induced by the hot SVA treatment, the EL spectra was dominated by the singlet emissions of PCE11 and PCBM, as depicted in Fig. 3a, which is in excellent agreement with the EL data collected from aged BHJ devices. As shown in Supplementary Fig. 8, the $J_{SC}$ of OSCs treated with SVA significantly decreased, but is comparable to that of the optimized OSCs aged at room temperature in the dark. The EL data clearly suggest a strong phase separation in the amorphous donor/acceptor mixed regimes, which was further confirmed by FTPS spectroscopy. As shown in Fig. 3b, the CT absorption of the fresh sample is 4–5 times stronger than that of the aged one, indicating demixing of the donor and acceptor phases in the BHJ micromorphology. The parameters used to fit the CT absorbance band by a Gaussian function are summarized in Supplementary Table 2. It is worth mentioning that the FTPS characteristics of the aged PCE11:PCBM sample does not show a clear CT feature, in comparison to the fresh sample. Moreover, the peak maximum of the EL spectrum remained unchanged at 1.39 eV for aged PCE11:PCBM, which might be due to the

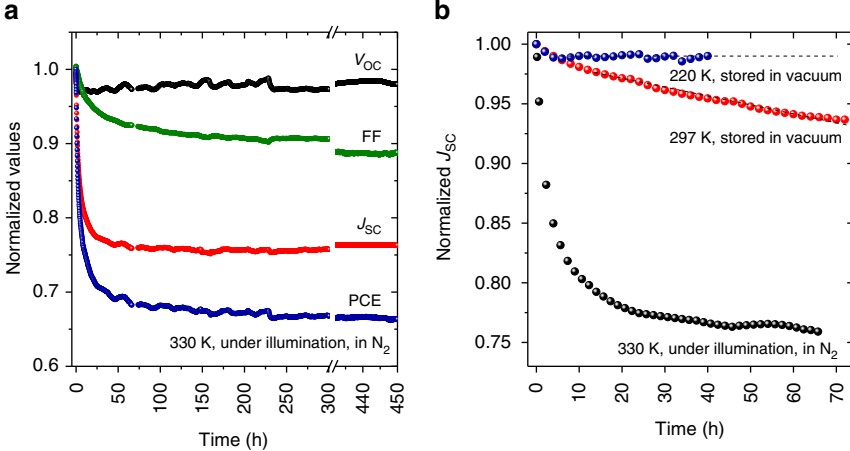

**Figure 2 | Stability test of PCE11:PCBM organic solar cells.** (**a**) Evolution of open-circuit voltage ($V_{OC}$), fill factor (FF), short-circuit current density ($J_{SC}$) and power conversion efficiency (PCE) of an optimized PCE11:PCBM solar cell measured under continuous 1 sun equivalent illumination (in $N_2$ atmosphere) for 450 h. (**b**) Evolution of $J_{SC}$ of optimized PCE11:PCBM solar cells measured at different temperatures.

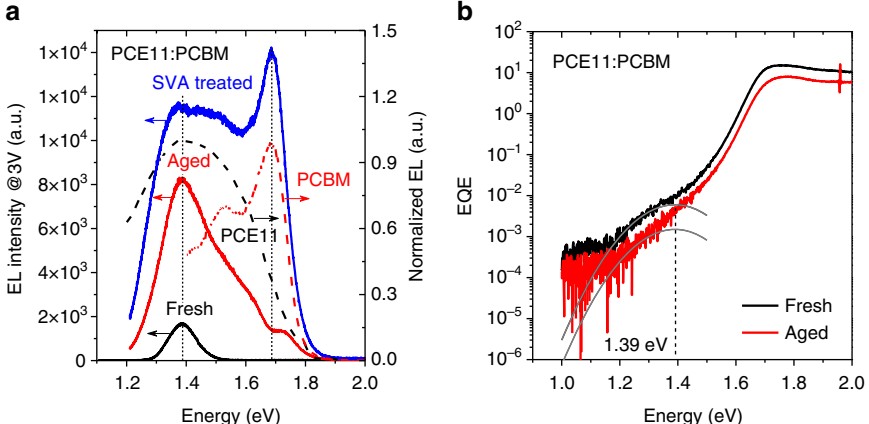

**Figure 3 | Electroluminescence (EL) and Fourier transform infrared spectroscopy (FTPS) spectra of PCE11:PCBM organic solar cells.** (**a**) EL spectra of PCE11:PCBM fresh, aged and solvent vapour annealing (SVA) treated samples. The EL spectra of PCE11 and PCBM neat materials are depicted for comparison. (**b**) External quantum efficiency (EQE) spectra of PCE11:PCBM fresh and aged samples measured by FTPS.

strongly overlapped EL emission with the neat PCE11. Therefore, although the centroid for aged PCE11:PCBM was fixed to 1.39 eV, the real width of CT absorbance cannot be determined precisely.

The donor/acceptor mixed region contributes to the exciton dissociation and generates free charge carriers in BHJ OSCs. Demixing of the donor and acceptor phases, which normally occurs upon thermal or solvent vapour treatment, may significantly reduce charge generation[4,13], and as a consequence, cause this abnormal strong burn-in loss in the PCE11:PCBM OSCs. The demixing of the donor and acceptor phases occurs only in the microscopic range and does not influence the UV-Vis-NIR absorption of OSCs, as shown in Supplementary Fig. 9 (refs 49,50). Six identical samples were prepared for grazing-incidence wide-angle X-ray scattering (GIWAXS) measurements and each sample was probed at three different positions to check the repeatability and reproducibility of the measurements. The 2D GIWAXS patterns and the corresponding azimuth information are shown in Supplementary Fig. 10. The GIWAXS profiles collected from out-of-plane cuts at $Q_y = 0$ and in-plane cuts at $Q_z = 0$ are depicted in Fig. 4a,b, respectively. The GIWAXS data of PCE11:PCBM fresh and aged samples confirm that the crystalline PCE11 and PCBM domains in the BHJ structure remain unchanged, indicating that the bi-continuous percolation pathways were not affected in the

aged OSCs. Consequently, the FF of the aged samples remained at a high level, as the photo-generated charge carriers can still be effectively collected through the percolation pathways formed in the BHJ OSCs.

We performed grazing-incidence small-angle X-ray scattering (GISAXS) measurements to analyse the morphological properties of the finely mixed amorphous regions[51,52]. The 2D GISAXS patterns of PCE11:PCBM and neat PCE11 samples are shown in Supplementary Fig. 11. The in-plane GISAXS profiles of fresh and aged neat PCE11 are almost identical as shown in Supplementary Fig. 12. On the other hand, fresh PCE11:PCBM shows a sharper shoulder or hump compared to the aged sample as depicted in Fig. 4c. The shoulder shape can be regarded as the intensities contributed mainly by the form factor scattering of the pure PCBM clusters/domains. Lorentz-correction of the scattered intensity as shown in Fig. 4d can be used to determine the position of this shoulder[51,53,54]. This model-independent analysis is self-consistent with the fitting of the GISAXS profiles using the combination of poly-dispersed spheres having a Schultz size distribution with the hard-sphere interaction between PCBM clusters and Debye−Anderson−Brumberger (DAB) model (Supplementary Table 3). DAB model characterizes the network of PCBM molecules distributed within the amorphous and around the crystalline polymer molecular conformations[55]. As

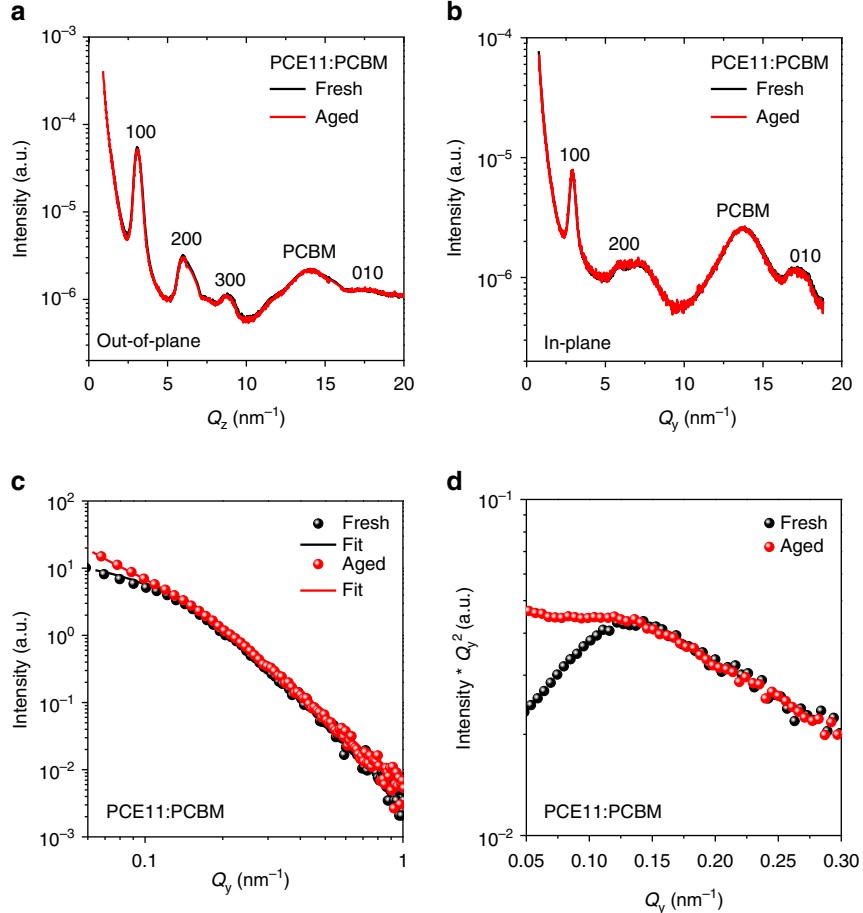

**Figure 4 | Grazing-incidence wide-angle X-ray scattering (GIWAXS) and grazing-incidence small-angle X-ray scattering (GISAXS) data of PCE11:PCBM samples.** The GIWAXS profiles of PCE11:PCBM fresh and aged samples collected from (**a**) out-of-plane cuts at $Q_y = 0$ and (**b**) in-plane cuts at $Q_z = 0$. The GISAXS profiles and model fitting of fresh and aged PCE11:PCBM samples collected from in-plane cuts made at an exit angle equal to the Yoneda peak of the active layer. The data are plotted in two forms: (**c**) scattering intensity I versus scattering vector $Q_y$ and (**d**) I × $Q_y^2$ versus $Q_y$ representing the Lorentz-corrected data.

the size of pure PCBM domains increases in the aged sample, their volume fraction and polydispersity of size distribution increase as well. This suggests that diffusion of fullerene molecules out of the finely mixed regions results in growth of larger clusters for the aged PCE11:PCBM sample.

To analyse the thermal behaviour and miscibility of the PCE11:PCBM BHJ micromorphology, differential scanning calorimetry (DSC) measurements were performed on various PCE11:PCBM blends[56,57]. Figure 5a depicts the DSC scans of neat PCE11, neat PCBM and PCE11:PCBM 1:1 blend. Melting peaks can be clearly seen from the heating scans of both neat materials, indicating the existence of crystalline phases in both neat materials. Two distinct melting peaks were found from the 1st heating scan of the PCE11:PCBM 1:1. The melting peak at 281 °C was composed by the crystallites of PCE11 and PCBM, while the one at 257 °C was composed by the mixed crystallites of PCE11:PCBM blend. Although both melting peaks were detected from the second heating scan of the PCE11:PCBM 1:1 blend, the nature of crystallites was changed upon the melting and cooling processes. With respect to the second heating scan of the PCE11:PCBM 1:1 blend, the melting peak at 285 °C was solely composed by the crystallites of PCBM, while the one at 254 °C was mainly composed by the imperfect crystallites of PCE11 mixed with PCBM. The melting and cooling enthalpy change

($\Delta H$) of PCE11:PCBM 1:1 blend and the corresponding neat materials are summarized in the Supplementary Table 4. As shown in Fig. 5b, by adding 15% PCE11 into the blend, the melting temperature of PCBM crystallites remained unaffected, while melting-point depression was observed for PCE11 crystallites by adding only 5% PCBM into the blend. The DSC data of the PCE11:PCBM blends revealed that the thermal behaviour of PCE11 is easily influenced by adding small amount of PCBM, forming imperfect crystallites in the blend. As shown in Supplementary Fig. 13, all the PCE11:PCBM blends exhibited similar well-defined crystallization peaks with the contribution mainly from PCE11. Although the reduction in $\Delta H$ for PCE11:PCBM blends can be attributed to impurities in PCE11 crystallites rising from PCBM addition, the formed PCE11:PCBM mixed crystallites exhibited comparable high $\Delta H$ when correction for the volume fraction of PCBM (Supplementary Table 5). Moreover, it is very important to point out that from the sample with 30% PCE11 loading the contribution from both PCBM and PCE11 crystallites can be well resolved (Supplementary Fig. 13). The $\Delta H$ of PCE11:PCBM with 30% PCE11 loading was determined to be $21.95 \, \mathrm{J \, g^{-1}}$, which is significantly higher than that of PCBM ($9.605 \, \mathrm{J \, g^{-1}}$), as well as the calculated value by simply taking into account the volume fraction of PCE11 and PCBM crystallites ($14.39 \, \mathrm{J \, g^{-1}}$). To summarize, the well-defined

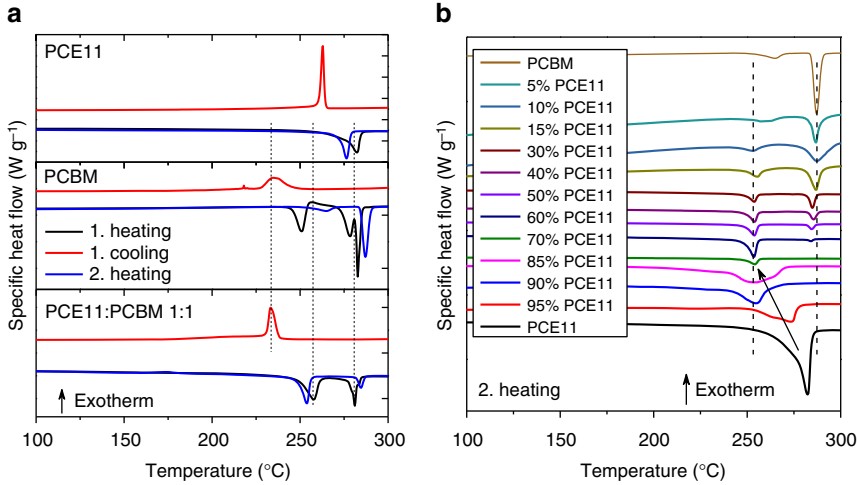

**Figure 5 | Differential scanning calorimetry (DSC) thermal analysis of PCE11:PCBM samples. (a)** DSC scans of neat PCE11, neat PCBM and PCE11:PCBM 1:1 blend. (**b**) The second DSC heating scans of PCE11:PCBM samples.

crystallization of PCE11 facilitates the formation of the mixed PCE11:PCBM crystallites, especially for the blends with high PCBM loadings.

**Polymer/fullerene liquid (melt) solid transition.** To underpin the relevance and the importance of the miscibility for the microstructural stability, a thermodynamic model based on *ab initio* density functional theory (DFT) quantum chemistry calculations named conductor-like screening model for real solvents (COSMO-RS) was employed to analyse the origin of the metastable PCE11:PCBM BHJ micromorphology. It is worth mentioning that an easy to use computational approach allowing to predict the microstructure stability of bulk-heterojunction composites would be of enormous value. The approach used for the calculation relies on the COSMO-RS allowing to determine the chemical potential of a molecule, which further represents the thermodynamic material properties and is detailed in the Supplementary Information. An overview of the theoretical calculations is depicted in Supplementary Fig. 14. The values of molecular weight and volume estimated for P3HT, PCE11 and PCBM using the high-quality quantum chemical *ab initio* electronic structure optimization are summarized in Supplementary Table 6. Experimental values of P3HT and PCBM reported in literature are in great agreement with the values estimated using the computational approach. It must be underlined that the molecular weight/volume of homo-polymers, such as P3HT, were estimated using their repeating units. However, owing to the complex chemical structures of D-A co-polymers, it is not reasonable anymore to estimate the molecular weight/volume according to the repeating units. To have a better comparison with P3HT, the molecular weight and volume of PCE11 were estimated according to the fraction of each component, as shown in Supplementary Fig. 15. The polymer/PCBM liquid (melt) solid transition can be determined to compare the phase behaviour between P3HT:PCBM and PCE11:PCBM. The condition $\partial^2 \Delta G_{mix}/\partial \varphi^2 > 0$ is required and the value of spinodal interaction parameter $\chi_{spinodal}$ that defines the boundary between the two-phase region and homogenous region can be derived[58–62]:

$$\chi_{spinodal} = \frac{v_0}{2}\left(\frac{\rho_1}{M_1\varphi_1} + \frac{\rho_2}{M_2(1-\varphi_1)}\right) \tag{1}$$

where $v_0$ is the molar volume of the lattice site in the Flory–Huggins model; $\rho_1$, $M_1$ and $\varphi_1$ are the density, the molecular weight and the fraction volume of PCBM, respectively;

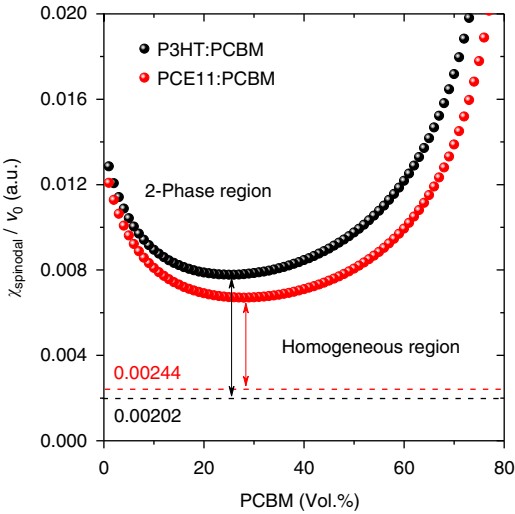

**Figure 6 | Polymer/PCBM liquid (melt) solid transition diagrams.** The polymer/PCBM liquid (melt) solid transition diagrams estimated for P3HT:PCBM and PCE11:PCBM as a function of the volume fraction of PCBM. The dashed lines represent the interaction parameters of polymer–fullerene blends.

$\rho_2$ and $M_2$ are the density and the molecular weight of polymer, respectively. The interaction parameters summarized in Supplementary Table 7 were calculated for P3HT:PCBM and PCE11:PCBM according to equation (2) from homogeneous Flory–Huggins solution theory[59,60]:

$$\chi_{1,2} = \frac{v_0}{RT}(\delta_{T_1} - \delta_{T_2})^2 \tag{2}$$

where $\chi_{1,2}$ is the polymer–fullerene interaction parameter, $v_0$ is the molar volume of the lattice site in the Flory–Huggins model, $\delta_T$ is the Hildebrand solubility parameter, which predicts that solvents and organic compounds with similar cohesive energy density would be more miscible. The entropic contribution is usually between $10^{-6}$ and $10^{-2}$, which is smaller in magnitude than the enthalpic contribution given for polymer solvent ($\sim 0.34$)[63–65]. Therefore, the entropic contribution was not taken into account for calculation.

Figure 6 depicts the polymer/PCBM liquid (melt) solid transition of the P3HT:PCBM and PCE11:PCBM systems as a

function of the volume fraction of PCBM. The use of the ratio of $\chi_{spinodal}$ to $\upsilon_0$ in Fig. 6 allows for a better comparison between the two systems. The liquid (melt) solid transition of PCE11:PCBM is overall lower than that of P3HT:PCBM, indicating that PCE11:PCBM is more prone to phase separation than P3HT:PCBM. The dashed lines represent the interaction parameters determined by theoretical calculations. Although the values are located in the homogenous region for both systems, the interaction parameter for PCE11:PCBM is much close to the spinodal demixing line compared to that of P3HT:PCBM, again underlining the metastable BHJ morphology of PCE11:PCBM blend. Even though advanced processing allows to form an intermixed micromorphology, in the case of PCE11:PCBM kinetics cannot overcome thermodynamics. The inherently low miscibility between PCE11 and PCBM leads to spontaneous phase separation in the solid state at room temperature, thus reducing the photovoltaic performance of the PCE11:PCBM OSCs.

## Discussion

This work demonstrates the demixing of polymer donor and fullerene acceptor phases at room temperature in state-of-the-art OSCs, and reports its significant influence on charge carriers generation as well as device performance. As the metastable micro-morphology of PCE11:PCBM is ascribed to the inherent low miscibility of PCE11 and PCBM, replacing the PCBM with other functional acceptors with good miscibility would be one of the most elegant ways to overcome this issue. It has been reported that indene-$C_{60}$ bisadduct (ICBA) shows better miscibility with P3HT than PCBM, and therefore is expected to be a promising alternative to PCBM for stable solar cells[59]. However, the PCE11:ICBA does not perform well compared to the PCE11:PCBM, as shown in Supplementary Fig. 16, and it is therefore not reasonable to experimentally verify the device stability. Nevertheless, we used the theoretical calculation discussed in the previous section to predict the miscibility and the corresponding stability of PCE11:ICBA blend. As the PCBM and ICBA have similar molecular weight and density, the values of $\chi_{spinodal}/\upsilon_0$ versus fraction volume are very similar for P3HT:PCBM and P3HT:ICBA, as well as for PCE11:PCBM and PCE11:ICBA. However, as depicted in Supplementary Fig. 17, the interaction parameter $\chi_{1,2}/\upsilon_0$ of P3HT:PCBM is more close to the spinodal demixing line than that of P3HT:ICBA, indicating that the P3HT:PCBM is expected to be less miscible than P3HT:ICBA. This theoretical calculation is in great agreement with the experimental results published in literature[59]. The same to the PCE11-based systems, PCE11:ICBA is expected to be more miscible than PCE11:PCBM as well. A recent publication by Cheng et al.[66] confirmed that the thermal stability of PCE11:PCBM can be significantly improved by adding small amounts of ICBA into the active blends, which agrees well with our theoretical prediction.

To summarize, we presented an abnormal strong photocurrent burn-in loss caused by demixing of the donor and acceptor phases in highly efficient PCE11:PCBM OSCs. The $J_{SC}$ of the optimized OSCs was reduced by 30–40% after 5 days stored in dark under ambient conditions. The spontaneous phase separation in the mixed amorphous regimes, which occurs at room temperature and is independent of the storage conditions, is attributed to an inherently low miscibility between PCE11 and PCBM. Therefore, the kinetically (that is, by advanced processing) optimized BHJ microstructure cannot be maintained even at room temperature for solid films. The demixing of the donor and acceptor mixed phases were confirmed by the EL, FTPS and GISAXS data, while GIWAXS data proved that the crystalline,

pure PCE11 and PCBM domains remained quite unchanged, which is in great accordance with the high FF observed for aged OSCs. Computational modelling in combination with thermodynamic calculations allowed to pin down the responsible mechanism. Although the backbone of PCE11 based on quarterthiophene is very similar to P3HT, the distinguished polymer/PCBM liquid (melt) solid transition diagrams between PCE11:PCBM and P3HT:PCBM suggest that the type of polymer side chains as well as repeating unit in the backbone do play a crucial role in determining the miscibility between donor and acceptor.

It is notable to mention that these findings are targeted to fundamentally show that spinodal demixing is a major issue correlating stability and lifetime. However, we are certainly aware that different molecular weight distributions of the same system and of course different fullerenes or different polymer side chains may strengthen or weaken the driving forces for spinodal demixing. Nevertheless, the significance of these findings goes wide beyond the material couple PCE11:PCBM. Our findings suggest that extended thermodynamic calculations allow to properly predict the stability of BHJ microstructures. This will offer a powerful tool for the design of materials with enhanced miscibility, but also will allow to derive more general design rules for long time stable donor–acceptor composites.

## Methods

**Materials.** PCE11 (PffBT4T-2OD) (batch: YY8074CB and YY8098CB) and PCBM (99.5%) were purchased from 1-Material and Solenne BV, respectively. ZnO nanoparticles dispersed in ethanol were received from Nanograde AG. All the materials were used as received without further purification.

**Fabrication of PCE11:PCBM OSCs.** The OSCs investigated were fabricated on ITO-coated glasses using doctor-blading under ambient conditions. The relative humidity in our clean room was well controlled in the range of 40–45%, and the temperature was controlled to $\sim 22\,^\circ$C. The substrate was cleaned by ultra-sonication in acetone and isopropanol for 10 min each, and blow-dried using a nitrogen gun. The substrate was coated with 30 nm thick ZnO nanoparticles and dried at 80 $^\circ$C for 5 min. The active layer with a thickness of 350 nm was bladed from a chlorobenzene (CB): dichlorobenzene (DCB) (1:1) mixed solution of PCE11:PCBM with a mixture ratio of 1:1.2 wt.% (33 mg mL$^{-1}$ in total). Three vol.-% DIO was added to the solution 1 hour prior to deposition. The temperature of PCE11:PCBM solution was kept at 120 $^\circ$C and the temperature of substrate was kept at 100 $^\circ$C during coating. The substrate was immediately removed from the 100 $^\circ$C hot plate when the active layer was dried. The solar cells were completed by thermal evaporation of 15 nm MoOx and 100 nm Ag at 10$^{-6}$ mbar.

**Characterizations.** All the J–V characteristics were recorded using a source measurement unit from BoTest. Illumination was provided by a solar simulator (Oriel Sol 1A, from Newport) with AM1.5G spectra at 100 mW cm$^{-2}$, which was calibrated by a certified silicon solar cell. The active area of the constructed OSCs were defined by the overlap of the bottom and top electrode, which was determined to be 10.4 mm$^2$ for the OSCs based on ITO-coated glass and vacuum-deposited top electrode. The thicknesses of the films were measured with a profilometer (Tencor Alpha Step D 100). The absorption spectra of the active layers were recorded by a UV-Vis-NIR spectrometer (Lambda 950, from Perkin Elmer).

The EL measurements were performed by applying an external voltage/current source through the devices that have an active area of 10.4 mm$^2$. The luminescence spectra were collected in a back-scattering geometry, dispersed by an iHR320 monochromator (Horiba Jobin-Yvon), and recorded with a Peltier-cooled Si CCD (Synapse, Horiba Jobin-Yvon). Temperature dependent J–V characterization was done inside a VNF-100 N$_2$-flow cryostat from the Janis Research Company and recorded by a Keithley 236 Source-Measure-Unit. The illumination was provided by a white LED (XLamp CXA2011 1300K CCT) and its intensity was adjusted in a way that the generated photocurrent matched the one under a solar simulator. The moisture and O$_2$ concentrations were kept below 0.5 p.p.m. for the ageing experiments in N$_2$ atmosphere. The FTPS measurements were carried out using a Vertex 70 from Brucker optics, equipped with QTH lamp, quartz beam splitter and external detector option. A low-noise current amplifier (DLPCA-200) was used to amplify the photocurrent produced upon illumination of the photovoltaic devices with light modulated by the FTIR. The output voltage of the current amplifier was fed back to the external detector port of the FTIR, in order to be able to use the FTIR's software to collect the photocurrent spectrum.

**GIWAXS/GISAXS measurements.** The GIWAXS/GISAXS patterns were collected with the highly customized Versatile Advanced X-ray Scattering instrumenT ERlangen (VAXSTER) at the chair for Crystallography and Structural Physics, FAU, Germany. The system is equipped with a MetalJet D2 70 kV X-ray source from EXCILLUM, Sweden. The beam was shaped by a150 mm Montel optics (INCOATEC, Geesthacht) and two of the available four double-slit systems with the last slit system equipped with low scattering blades (JJXray/SAXSLAB). Aperture sizes were $(0.7 \times 0.7\,mm^2, 0.4 \times 0.4\,mm^2)$ for GIWAXS and $(0.15 \times 0.06\,mm^2, 0.12 \times 0.048\,mm^2)$ for GISAXS. The sample position was located within the fully evacuated detector tube. The hybrid-pixel 2D Pilatus 300 K detector (Dectris Ltd., Baden, Switzerland) was used to collect the scattered radiation. The measurements were carried out at energy of 9.28 keV. The collimation line was tilted and shifted with respect to the horizontal plane allowing grazing incidence angles which maximize the scattering volume and enhance the scattered intensity. The incidence angle α for GIWAXS measurements was between 0.183 and 0.189°, which is smaller than the critical angle of total reflection of the glass substrate and ZnO layer to limit the penetration depth and the scattering to the thin layer. Grazing incidence geometry of the incident X-ray with respect to the sample surface is used here to enhance the scattered intensity, to maximize the scattering volume, and to access the three dimensional (3D) structure of the studied thin films (lateral and normal direction). The detector-to-sample distance (SDD) was calibrated with a silver behenate standard to 172.7 mm for GIWAXS and 1,542.6 mm for GISAXS. Active layers were bladed on Si substrates for GISAXS measurements to avoid the intense contribution from ZnO to the scattering at low Q values. An incidence angle α of 0.25° was chosen to obtain a clear separation between the Yoneda peaks of the involved materials and the specular peak in GISAXS. Data were reduced with dpdak software[67]. The structural model for the reduced GISAXS data was fitted using the program SASfit[68].

**Theoretical calculation of molecular parameters.** The 3D molecular structures were built using HyperChem Professional7, which calculates the most stable conformational structures under the force field method. The raw 3D structure was exported as .xyz file format to TURBOMOLE6.3 quantum chemical program package for their molecular optimization by using high-quality quantum chemical ab initio electronic structure optimization with Ahlrichs triple zeta valence polarization basis set and the Becke Perdew DFT in the conductor like screen model (BP-TSVP-COSMO) quantum basis to create the COSMO file. Subsequently, its COSMO file was used as input in COSMOtherm14 statistical thermodynamic code to calculate the specific molecular weight (Mol. weight (g mol_1)), the molar volume (Mol. volume (cm3 mol_1)) and the corresponding density of the molecules at 298K (ref. 69). According to the chosen quantum method, the functional and the basis set, we used the corresponding parameterization BPTZVP-C21-1010 that is required for the calculation of physicochemical data and contains intrinsic parameters of COSMOtherm14 as well as element-specific parameters.

**DSC measurements.** PCE11 and PCBM were dissolved in CB:DCB (1:1) at a concentration of 10 mg ml$^{-1}$ and stirred in a glovebox at 80 °C for overnight. The solutions were mixed with different PCE11:PCBM ratios and again stirred at 80 °C for overnight. The homogeneous solutions were dried on clean substrates in a glovebox at 80 °C. For each measurement, 3–5 mg material was collected from the substrate and filled into a DSC pan. DSC measurements were carried out using a Q1000 DSC setup from TA Instruments. The temperature of conventional DSC measurements ranges from −50 °C to 310 °C with a heating and cooling rate of 10 K min$^{-1}$. For the modulated DSC measurement, the temperature ranges from −50 °C to 310 °C with a heating and cooling rate of 3 K min$^{-1}$ and a modulated temperature of ±1 K min$^{-1}$.

**Data availability.** The data that support the findings of this study are available from the corresponding author upon request.

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

## Acknowledgements

N.L. acknowledges the financial support by the Bavarian Ministry of Economic Affairs and Media, Energy and Technology for the joint projects in the framework of the Helmholtz Institute Erlangen-Nürnberg for Renewable Energy (IEK-11) of Forschungszentrum Jülich, and the financial support from the Emerging Talents Initiative (ETI) at FAU Erlangen-Nürnberg. J.D.P. is funded by a doctoral fellowship grant of the Colombian Agency COLCIENCIAS. Y.H. is grateful for the funding of the Erlangen Graduate School in Advanced Optical Technologies (SAOT) at FAU Erlangen-Nürnberg. H.C. and S.C. acknowledge the financial support from the China Scholarship Council (CSC). C.J.B. gratefully acknowledges the financial support through the 'Aufbruch Bayern' initiative of the state of Bavaria (EnCN and solar factory of the future), the Bavarian Initiative 'Solar Technologies go Hybrid' (SolTech), the SFB 953 Synthetic Carbon allotropes and the Cluster of Excellence 'Engineering of Advanced Materials' (EAM) FAU Erlangen-Nürnberg. T.U., M.B. and T.K. acknowledge the support by the Cluster of Excellence 'Engineering of Advanced Materials' (EAM), CENEM, and GRK 1896. S.L. is funded by the project 'Umweltfreundliche hocheffiziente organische Solarzellen' (UOS).

## Author contributions

N.L. and C.J.B. conceived the idea and directed the project. N.L. coordinated the experiments, performed device fabrication and characterization, the DSC measurements and analysed the data. M.R. and T.H. conducted the stability test. J.D.P. and S.L. performed the theoretical calculation. G.J.M. conducted the FTPS measurements. N.S.G. and H.C. performed the EL measurements. Y.H. and S.C. assisted with device fabrication and data interpretation. T.K., M.B. and T.U. performed the GIWAXS and GISAXS measurements and analysed the data. N.L. and C.J.B. wrote the manuscript with input from J.D.P. and T.K.

## Additional information

**Competing financial interests**: The authors declare no competing financial interests.

**How to cite this article**: Li, N. *et al.* Abnormal strong burn-in degradation of highly efficient polymer solar cells caused by spinodal donor-acceptor demixing. *Nat. Commun.* **8,** 14541 doi: 10.1038/ncomms14541 (2017).

**Publisher's note**: 

