## [Peer Review File · Nature Communications]

Reviewers' comments:

Reviewer #1 (Remarks to the Author):

Decision: Reject in present format. Major revisions required prior to resubmission, otherwise will not meet publication guidelines for Nature Communications.

The authors report an apparently abnormal burn-in phase in a novel bulk heterojunction organic photovoltaic (OPV). They propose that this process is caused by a thermodynamically unstable active layer nanostructure. Specifically, under ambient conditions the conjugated copolymer and fullerene undergo excessive demixing over several days via spinodal decomposition. As a consequence, the interfacial volume between both materials is reduced and the efficiency of free charge generation suffers. This lowers the achievable photocurrent density of the solar cell by ~ 30 - 40% from its initial value. The authors claim that such a process, which has not been reported to date, must be taken into account when designing organic photovoltaic cells that offer high power conversion efficiency (PCE) and long-term stability.

The research is certainly topical and has the potential to generate strong impact in the field. Given the complex nature of failure modes in OPVs I am willing to accept that unstable active layer nanostructures will eventually contribute to a reduced PCE over time. Unfortunately there is insufficient evidence in the manuscript to support such a specific claim over the timescales investigated. I am also surprised by the decision to fabricate the OPVs in ambient conditions prior to ageing as this will inevitably introduce additional factors into the stability investigation. Given the well known sensitivity of organic semiconductors to water and oxygen (e.g. doping, accelerated trap formation and photobleaching), fabricating OPVs in air is likely to cause unnecessary damage to the semiconductors. How do the authors know the results obtained aren't simply due to air exposure during device fabrication?

To simplify the experiment the effects of an unstable active layer nanostructure (if genuine) must be isolated from environmental factors. Control solar cells should be prepared under standard glovebox conditions, encapsulated, measured, and aged under dry N₂/dark conditions prior to the second measurement. If the PCE changes by a significant amount this can be studied using standard techniques. In its present format, the study is simply unconvincing and contains too much uncertainty to confidently ascribe the changes in OPV PCE to a single degradation mechanism.

Other aspects of the study that should be revised include:

1 - Confirming a modified active layer post-ageing. The authors should provide direct structural evidence of this. If the relative volume fractions and electron densities of amorphous and mixed-amorphous domains in the sample changes this should be resolvable in a small angle X-ray scattering experiment (see DOI: 10.1021/ma2007706 and DOI: 10.1002/aenm.201301377 for example data) or similar.

2 - Semiconductor properties related to a modified active layer nanostructure. The uv-vis data in the supporting information (S.I.) should be presented on an absolute rather than a normalised scale so that the reader can see if any loss in absorption has taken place. The parameters used to model the CT region of the EQE spectrum should be stated and discussed. It would be interesting to see how much of a change ageing induces in the CT absorbance band width as this is typically seen as a measure of disorder at the donor:acceptor interface.

3 - Ageing under continuous illumination. The spectrum of the LED used in these experiments should be provided in the S.I. The glass transition temperature (T_g) of the PCE11:PCBM thin-film should be measured and discussed in the context of the 350K ageing data (is the rapid drop in PCE a result of being at T_g or above it?). The moisture content of the N₂ atmosphere used during

certain ageing experiments should be stated.

4 - In the methods section, the following details are missing: the relative humidity during OPV active layer deposition and the amount of time spent by the substrate at 100C (i.e. during spin coating).

5 - In the introduction, there are limited references on the stability of state-of-the-art OPVs, let alone on older systems (e.g. P3HT:PCBM, PCDTBT:PC70BM). These highly relevant studies should be referenced to help put the 'abnormal strong burn-in' into context. On line 132, the following papers should be cited to help the reader appreciate that diiodooctane (DIO) may negatively affect the performance/stability figure of merit for a wide range of OPVs, not just those prepared under ambient conditions: DOI: 10.1021/jp510996w, doi:10.1016/j.orgel.2015.12.024.

Although I do not recommend publication at this time, the research should be developed as short term stability in OPVs is a pressing challenge and strategies to solve this are sorely lacking. Thoughtful and careful redesign of the core experiments should go some way to help address this.

Reviewer #2 (Remarks to the Author):

The manuscript 'Abnormal strong burn-in degradation of highly efficient polymer solar cells caused by spinodal donor-acceptor demixing' provides striking evidence of the fast degradation the high-efficiency OPV blends based on PCE11 as donor material and fullerenes such as PCBM-C60 undergo. The manuscript clearly demonstrates that one should not only seek to obtain high efficiencies and highlights unambiguously what issues in materials- and device design need to be overcome if OPVs become a commercially viable technology. Already for this message, delivered in such a convincing way, I recommend the manuscript for publication in Nature Communications. A broad audience has to receive this information.

The strength of the message of the present manuscripts stems from the very conclusive device data, supported by EL (PL) data of blends and neat (not pristine!) materials. The emission data clearly indicates phase separation, as suggested by the authors. Some other parts are, however, based on data that are difficult to interpret (or should maybe be interpreted more carefully). This would make the manuscript more accessible for a broad audience. I have no doubts that the authors can do so and I suggest to give them this chance; I would like to see this manuscript be published in Nat. Comms. It deserves it!

Points to address:

1) UV-vis data: the authors indirectly use the Spano model and attempt to deduce information on the structural order from their UV-vis data. A) References to work by Spano and co-workers should be provided (more on reference below). B) The authors may read some of the recent work by Spano. The spectra their blends and neat PCE11 display are more typical for J-like aggregates. Hence, the fact that the spectra do not change, only implies that the electronic coupling between chains stays weak and there is a strong coupling along the polymer backbones. This may be due to the more rigid backbone of PCE 11 compared to P3HT; although it cannot drastically more rigid as the melting temperatures are comparable. In P3HT the 0-0 transition often changes as it is usually more H-like with strong electronic coupling perpendicular to the backbone. Hence, generally, a more H-like behaviour infers better electronic coupling along the pi-stack and, as a consequence, can be used as an INDICATION of improved order at least on short length scales. It not necessarily affects the microscopic range, as suggested on page 9. I suggest the authors have a close look at this section.

2) X-ray data. PCE11 features rather strong reflections; hence, changes on smaller length scales

may not be noted even in a log intensity plot. Indeed, I am not sure whether the fact that they see no changes really implies no change in molecular order or whether these changes are hidden in the 'amorphous' background. Because of the reasoning on the interpretation of the presented UV-vis data, there still could be changes on short ranges. I would combine here some of the DSC data the authors must have already, I am sure. Some details need to be given,: A) How were the DSCs measured on films prepared from solution or powders that simply were mixed? I may have missed the details; I have doubled checked the SI for it. In case the DSCs were made from cast films, it is important to use the first heating scan as compatibilisation effects through the solvent can lead to another phase behaviour (the 2nd heating provides information for melt processed material, hence, is not that relevant; it can provide information on degradation, solvent compatibilisation effects when compared to the 1st scan, etc.). For obtaining information on the molecular order, the authors can compare the enthalpy of fusion for the various blends (1st heating) normalised to the polymer fraction. I certainly would also show the data for compositions in the 1:1 range (see also below). In addition, with the neat polymer featuring such a small supercooling and well defined crystallisation peak in the 1st cooling in the neat polymer, I would suggest to analyse what occurs with this feature upon blending and casting for all the blends. Does the supercooling increase (which means ordering starts to be hindered). Is its enthalpy getting reduced, etc.?

3) The above is important as I think the authors refer to spinodal donor-acceptor demixing as they like to suggest that demixing of the two materials occurs only in the amorphous phase. This is not that clearly spelled out in the manuscript – if the authors want to stick to this hypothesis. In my opinion, it may be, but it may not; based on the data provided I am not convinced. It still could be order-induced phase separation – just molecular ordering on smaller length scales than e.g. accessible with X-ray diffraction. I think it will be difficult to obtain data that solves this issue fully. I am not sure if it is needed at this stage. The burn-in is clear and I fully agree that it originates from phase separation. I personally would rephrase this part and may take out the word 'spinodal': i.e. I would suggest to simply go for 'Abnormal strong burn-in degradation of highly efficient polymer solar cells caused by donor-acceptor blends.' I know some calculations are presented supporting the spinodal idea, however, I wonder how much the graph presented in Figure 4 changes when slightly different parameters are used?

Some additional comments:

1) The thermal analysis data presented in Fig S7b indicates that PCE11:PCBM features a eutectic phase diagramme as the P3HT:PCBM binary does, with a eutectic point between 85 and 90 % PCE11 (so, at a much higher donor content than for P3HT:PCBM system). This phase behaviour implies that both melting points are depressed; the one of PCE11 seems more strongly depressed as the eutectic point is so much shifted towards the polymer rich side. Eutectic temperatures is around 250 C, hence the shoulder on the higher-temperature side for the 85% blend indicates that this feature is the PCBM melting – i.e. in my opinion also the PCBM melting is depressed. Clearly, having some more data for the compositions between 15% to 85 % will assist to identify correctly the eutectic composition. This will have significance whether it is or not spinodal decomposition.

2) I do not want to suggest even more experiments, but out of curiosity, have the authors performed measurements with ICBA with a higher T_g and it seems somewhat better miscibility with P3HT?

3) Finally, I think the authors should include some important references: on mixing, initial work from the Ade/McNeill groups, Treat/Chabinyk et al., Westacott/Stingelin, Russell group at UMASS, the group of Rasmus Schroeder in Germany; on spinodal decomposition of OPV blends: Steiner/Friend; phase diagrammes: Nelson/Stingelin, used also by Hadziiaonnou. UV-vis: Work by Spano et al.

In summary, I think the present work is intriguing and a broad audience will be interested in it and will learn from it. Hence, I strongly suggest publication – after issues pointed to above have been

clarified or addressed.

Reviewer #3 (Remarks to the Author):

The author demonstrated strong burn-in degradation in highly-efficient state-of-the-art OSCs induced by spinodal demixing of the donor and acceptor phases, which dramatically reduces charge generation and can be attributed to the inherently low miscibility of both materials. In my opinion, the topic of "stability of OSCs" is very important but the novelty of this paper is not enough for Nature Communications. I think Scientific Reports is a good choice for this paper. Some comments are shown as below:

1. We already know the reason of this degradation (metastable state of morphology), but we don't know how to overcome it. That's the point!
2. There is only one high-efficiency polymer/fullerene pair discussed in this paper, it's not enough to make a strong conclusion based on one case.
3. After the degradation of OSCs, the VOC always higher. Please add some discussion.

A point-by-point Response to Reviewers' Comments:

Referee: 1

Comments to the Author

The authors report an apparently abnormal burn-in phase in a novel bulk heterojunction organic photovoltaic (OPV). They propose that this process is caused by a thermodynamically unstable active layer nanostructure. Specifically, under ambient conditions the conjugated copolymer and fullerene undergo excessive demixing over several days via spinodal decomposition. As a consequence, the interfacial volume between both materials is reduced and the efficiency of free charge generation suffers. This lowers the achievable photocurrent density of the solar cell by $\sim 30 - 40\%$ from its initial value. The authors claim that such a process, which has not been reported to date, must be taken into account when designing organic photovoltaic cells that offer high power conversion efficiency (PCE) and long-term stability.

The research is certainly topical and has the potential to generate strong impact in the field. Given the complex nature of failure modes in OPVs I am willing to accept that unstable active layer nanostructures will eventually contribute to a reduced PCE over time. Unfortunately there is insufficient evidence in the manuscript to support such a specific claim over the timescales investigated. I am also surprised by the decision to fabricate the OPVs in ambient conditions prior to ageing as this will inevitably introduce additional factors into the stability investigation. Given the well known sensitivity of organic semiconductors to water and oxygen (e.g. doping, accelerated trap formation and photobleaching), fabricating OPVs in air is likely to cause unnecessary damage to the semiconductors. How do the authors know the results obtained aren't simply due to air exposure during device fabrication?

To simplify the experiment the effects of an unstable active layer nanostructure (if genuine) must be isolated from environmental factors. Control solar cells should be prepared under standard glovebox conditions, encapsulated, measured, and aged under dry N_2 /dark conditions prior to the second measurement. If the PCE changes by a significant amount this can be studied using standard techniques. In its present format, the study is simply unconvincing and contains too much uncertainty to confidently ascribe the changes in OPV PCE to a single degradation mechanism.

- We greatly thank the reviewer for the very positive comments on our work, and fully agree with the reviewer on this statement. Actually, before starting with the air-processed PCE11:PCBM solar cells, we firstly optimized the PCE11:PCBM solar cells in a glove box without exposure to the air, and characterized their stability under various conditions. The statistic photovoltaic parameters of the optimized PCE11:PCBM fabricated by spin-coating in N_2 atmosphere are summarized in **Figure R1a**. As shown in the **Figure R1b**, the solar cells without exposure to the air showed the same strong burn-in loss under continuous one sun illumination. Moreover, similar drop in J_{SC} was also observed for these devices stored in a glove box at room temperature in the dark. According to the experimental data we can conclude that the abnormal strong burn-in loss observed for the PCE11:PCBM system is not affected by the ambient conditions during device fabrication.

Figure R1 (a) statistic photovoltaic parameters of optimized PCE11:PCBM fabricated by spin-coated in N_2 atmosphere. (b) Evolution of photovoltaic parameters of an optimized PCE11:PCBM solar cell measured under continuous 1 sun illumination. The devices fabricated by spin-coating in nitrogen were treated without exposure to air.

Other aspects of the study that should be revised include:

- 1 - Confirming a resolvable in a small angle X-ray scattering experiment (see DOI: 10.1021/ma2007706 and DOI: 10.1002/aenm.201301377 for example data) or similar.
- We thank the reviewer for pointing out the pioneer works published in literature. The related references were added to the manuscript. As suggested by the reviewer, we performed the GISAXS measurements on the fresh and aged PCE11:PCBM sample to analyze the morphology change especially in the amorphous mixed region.

Figure R2 2D GISAXS patterns of (a) fresh PCE11:PCBM, (b) aged PCE11:PCBM, (c) fresh pure PCE11, and (d) aged pure PCE11 samples.

Figure R3 The GISAXS profiles of PCE11:PCBM and neat PCE11 fresh and aged samples collected from in-plane cuts made at an exit angle equal to the Yoneda peak of the active layer.

Figure R4 The Lorentz-corrected intensity profiles of PCE11:PCBM fresh and aged samples collected from in-plane cuts made at an exit angle equal to the Yoneda peak of the active layer.

The GISAXS data confirm the diffusion of fullerene molecules out of the finely-mixed regions. The in-plane GISAXS profiles of fresh and aged pure PCE11 are almost identical as shown in **Figure R2**. On the other hand, fresh PCE11:PCBM shows a sharper shoulder or hump compared to the aged sample as depicted in **Figure R3a**. The shoulder shape can be regarded as the intensities contributed mainly by the form factor scattering of the pure PCBM clusters/domains. Lorentz-correction of the scattered intensity can be used to determine the position of this shoulder, as shown in **Figure R4**.^{1,2} This model-independent analysis is self-consistent with the fitting of the GISAXS profiles using the combination of poly-dispersed spheres having a Schultz size distribution with the hard-sphere interaction between PCBM clusters and Debye–Anderson–Brumberger (DAB) model (cf. **Table R1**). DAB model characterizes the network of PCBM molecules distributed within the amorphous and around the crystalline polymer molecular conformations.³ The aged sample has a larger correlation length ζ within the DAB model and confirms an increased separation between PCBM molecules and the polymer crystallites. As the size of pure PCBM domains increases in the aged sample, their volume fraction and polydispersity of size distribution increase as well. This suggests that diffusion of fullerene molecules out of the finely-mixed regions results in growth of larger clusters for the aged PCE11:PCBM sample.

Table R1 Structural parameters determined by model-fitting of GISAXS profiles of PCE11:PCBM thin films.

	Correlation length of	Fullerene clusters
--	-----------------------	--------------------

	the mixed region (nm)	Average domain size (nm)	Width of size distribution (nm)	Volume fraction %
As fabricated	7.71109	43	0.208399	6.87
Aged in air	8.16728	78	0.872295	36.8322

2 - Semiconductor properties related to a modified active layer nanostructure. The uv-vis data in the supporting information (S.I.) should be presented on an absolute rather than a normalised scale so that the reader can see if any loss in absorption has taken place. The parameters used to model the CT region of the EQE spectrum should be stated and discussed. It would be interesting to see how much of a change ageing induces in the CT absorbance band width as this is typically seen as a measure of disorder at the donor:acceptor interface.

- The uv-vis spectra of fresh and aged PCE11:PCBM samples were taken from the entire device without top electrode, and normalized only to correct the baseline (not normalized from 0 to 1). As shown in the following figure, no degradation or loss was observed from the absorption spectra. The title of y-axis in **Figure R5** was corrected to Absorbance (a.u.).

Figure R5 UV-Vis absorption spectra of fresh and aged PCE:PCBM films.

The CT region of PCE11:PCBM was fitted with the GaussAMP function, which is shown in the following equation.

$$y = Ae^{-\frac{(x-x_c)}{2w^2}}$$

where A is the amplitude, x_c is the centroid and w is the width of the fitting. The fitting parameters used to model the CT region of the PCE11:PCBM blend are summarized in **Table R2**. The value of x_c was taken from the peak maximum of EL spectra, which is located at ~ 1.39 eV, as shown in **Figure R6**.

Table R2 Fitting parameters used for modeling the CT absorbance band of fresh and aged PCE11:PCBM.

	Fresh PCE11:PCBM	Aged PCE11:PCBM
x_c	1.39	1.39
w	0.1	0.1
A	0.006	0.0015

Figure R6 (a) EL spectra of PCE11:PCBM fresh and aged samples. The EL spectra of PCE11 and PCBM (PL) pristine materials are depicted for comparison. (b) EQE spectra of PCE11:PCBM fresh and aged samples measured by FTPS.

We fully agree with the reviewer that the change in CT absorbance band width is related to the change of disorder at the donor-acceptor interfaces. However, the aged BHJ sample is difficult to deconvolute as the FTPS characteristics of the aged PCE11:PCBM sample does not show a clear CT feature, in comparison to the fresh sample. Moreover, the peak maximum of the EL spectrum remained unchanged at ~ 1.39 eV for aged PCE11:PCBM,

which might be due to the strongly overlapped EL emission with the neat PCE11. Therefore, although E_c for aged PCE11:PCBM was fixed to 1.39 eV, the real width of CT absorbance cannot be determined precisely. We summarize the findings that the rather small changes in the CT population are quite small compared to the dramatic increase in the pristine polymer emission. We suggest that CT spectroscopy is probably not ideal to characterize PCE11:PCBM blends as the position of the CT absorption is very close to the polymer's bandgap.

3 - Ageing under continuous illumination. The spectrum of the LED used in these experiments should be provided in the S.I. The glass transition temperature (T_g) of the PCE11:PCBM thin-film should be measured and discussed in the context of the 350K ageing data (is the rapid drop in PCE a result of being at T_g or above it?). The moisture content of the N₂ atmosphere used during certain ageing experiments should be stated.

- The white LED (XLamp CXA2011 3000K CCT) used for the stability test was purchased from Cree Inc. The spectrum of the LED, shown in the following Figure, was added to the Supplementary **Figure R7**.

Figure R7 Illumination spectrum of the white LED used for the stability test.

As suggested by the reviewer, temperature modulated differential scanning calorimetry (m-DSC) measurements were performed to determine the glass transition temperature (T_g) of PCE11:PCBM 1:1 blend. The 2nd heating scan of PCE11, PCBM and PCE11:PCBM 1:1 blend are summarized in **Figure R8**. As depicted in the Figure, the T_g

of PCE11:PCBM 1:1 was determined to be 14.6 °C from the 2nd heating scan of m-DSC measurement, while no clear T_g was observed for neat PCE11 and neat PCBM.

Figure R8 The 2nd heating scans of PCE11, PCBM and PCE11:PCBM 1:1 blend measured by temperature modulated DSC.

The H₂O and O₂ content of the N₂ atmosphere are monitored by a hygrometer and an oxygen meter, respectively. Both H₂O and O₂ contents were kept below 0.5 ppm for the ageing experiments. As depicted in the picture, the dew point temperature and the O₂ concentration were determined to be -82.3°C and 0.33 ppm, respectively. Moreover, the ambient temperature during continuous one sun illumination was rechecked and corrected to ~330 K in the manuscript.

Figure R9 Picture of hygrometer and oxygen analyzer used for monitoring the moisture and oxygen concentration.

4 - In the methods section, the following details are missing: the relative humidity during OPV active layer deposition and the amount of time spent by the substrate at 100°C (i.e. during spin coating).

- The solar cells studied in this work were fabricated by doctor blading under ambient conditions in a clean room. The relative humidity in our clean room was well controlled in the range of 40-45%, and the temperature was controlled to ~22°C. The substrate was kept at 100°C prior to deposition of the active layer, and was immediately removed from the hot plate when the active layer was dried. The active layer was dried at 100°C less than 5 s. The solar cells treated with hot solvent vapor annealing, as shown in Figure XXX, were kept in a DCB atmosphere at 100°C for 20 min to intentionally create phase separation.

5 - In the introduction, there are limited references on the stability of state-of-the-art OPVs, let alone on older systems (e.g. P3HT:PCBM, PCDTBT:PC70BM). These highly relevant studies should be referenced to help put the 'abnormal strong burn-in' into context. On line 132, the following papers should be cited to help the reader appreciate that diiodooctane (DIO) may negatively affect the performance/stability figure of merit for a wide range of OPVs, not just those prepared under ambient conditions: DOI: 10.1021/jp510996w, doi:10.1016/j.orgel.2015.12.024.

- We thank the reviewer for pointing out the pioneer works. The related references have been added to the manuscript.

Although I do not recommend publication at this time, the research should be developed as short term stability in OPVs is a pressing challenge and strategies to solve this are sorely lacking. Thoughtful and careful redesign of the core experiments should go some way to help address this.

- We added a paragraph to the Discussion section to demonstrate the powerful tool based on the theoretical calculations for future design and development of novel organic material systems with promising miscibility. An alternative acceptor, such as ICBA or non-fullerene acceptor, would solve the inherent low miscibility of PCBM with some highly efficient polymers. It is hoped that the findings demonstrated in this work will trigger more research interests on this topic.

Referee: 2

Comments to the Author

The manuscript ‘Abnormal strong burn-in degradation of highly efficient polymer solar cells caused by spinodal donor-acceptor demixing’ provides striking evidence of the fast degradation the high-efficiency OPV blends based on PCE11 as donor material and fullerenes such as PCBM-C60 undergo. The manuscript clearly demonstrates that one should not only seek to obtain high efficiencies and highlights unambiguously what issues in materials- and device design need to be overcome if OPVs become a commercially viable technology. Already for this message, delivered in such a convincing way, I recommend the manuscript for publication in Nature Communications. A broad audience has to receive this information.

The strength of the message of the present manuscripts stems from the very conclusive device data, supported by EL (PL) data of blends and neat (not pristine!) materials. The emission data clearly indicates phase separation, as suggested by the authors. Some other parts are, however, based on data that are difficult to interpret (or should maybe be interpreted more carefully). This would make the manuscript more accessible for a broad audience. I have no doubts that the authors can do so and I suggest to give them this chance; I would like to see this manuscript be published in Nat. Comms. It deserves it!

- We greatly thank the reviewer for the very positive comments.

Points to address:

- 1) UV-vis data: the authors indirectly use the Spano model and attempt to deduce information on the structural order from their UV-vis data. A) References to work by Spano and co-workers should be provided (more on reference below). B) The authors may read some of the recent work by Spano. The spectra their blends and neat PCE11 display are more typical for J-like aggregates. Hence, the fact that the spectra do not change, only implies that the electronic coupling between chains stays weak and there is a strong coupling along the polymer backbones. This may be due to the more rigid backbone of PCE 11 compared to P3HT; although it cannot drastically more rigid as the melting temperatures are comparable. In P3HT the 0-0 transition often changes as it is usually more H-like with strong electronic coupling perpendicular to the backbone. Hence, generally, a more H-like behaviour infers better electronic coupling along the pi-stack and, as a consequence, can be used as an INDICATION of improved order at least

on short length scales. It not necessarily affects the microscopic range, as suggested on page 9. I suggest the authors have a close look at this section.

- We greatly thank the reviewer for the helpful suggestion and comments. The related works published by Spano et al. were added to the manuscript as Ref 50, 51. We fully agree that the backbone of PCE11 is more rigid than that of P3HT, and the crystallinity of neat PCE11 is also higher than that of P3HT when comparing the enthalpy change of the melting peak. The characterization results shown in this work directly or indirectly imply that the drastic reduction in J_{SC} for aged PCE11:PCBM solar cells is ascribed to the demixing of the donor and acceptor phases, even at room temperature in the dark. This is one of the core messages delivered by this contribution. The UV-vis data demonstrated in this work on the one hand revealed that polymer degradation does not occur which may also lead to the reduction in J_{SC} ; on the other hand confirmed along with the water contact angle measurement that the macroscopic morphology of BHJ film did not change. Together with the EL spectra, the FTPS data, the GIWAXS and the supplemented GISAXS data, we can come to a very clear conclusion that the drastic reduction in J_{SC} for aged PCE11:PCBM solar cells is indeed due to the abnormal unstable mixed donor-acceptor amorphous regions.

2) X-ray data. PCE11 features rather strong reflections; hence, changes on smaller length scales may not be noted even in a log intensity plot. Indeed, I am not sure whether the fact that they see no changes really implies no change in molecular order or whether these changes are hidden in the ‘amorphous’ background. Because of the reasoning on the interpretation of the presented UV-vis data, there still could be changes on short ranges. I would combine here some of the DSC data the authors must have already, I am sure. Some details need to be given,: A) How were the DSCs measured on films prepared from solution or powders that simply were mixed? I may have missed the details; I have doubled checked the SI for it. In case the DSCs were made from cast films, it is important to use the first heating scan as compatibilisation effects through the solvent can lead to another phase behaviour (the 2nd heating provides information for melt processed material, hence, is not that relevant; it can provide information on degradation, solvent compatibilisation effects when compared to the 1st scan, etc.). For obtaining information on the molecular order, the authors can compare the enthalpy of fusion for the various

blends (1st heating) normalised to the polymer fraction. I certainly would also show the data for compositions in the 1:1 range (see also below).

- To address the reviewer's concerns, GISAXS measurements were carried out on the fresh and aged PCE11:PCBM samples. The fresh and aged neat PCE11 samples were measured as references. As depicted in **Figures R2** and **R3**, the GISAXS data on fresh and aged PCE11:PCBM samples clearly confirm the demixing of the amorphous donor and acceptor phases. **Figure R4** shows that fresh and aged samples have a peak position corresponding to a domain size of about 44 and 77 nm, respectively. This model-independent analysis is self-consistent with the fitting of GISAXS profiles using the combination of poly-dispersed spheres having a Schultz size distribution with the hard-sphere interaction between PCBM clusters and Debye–Anderson–Brumberger (DAB) model, see **Table R1**.

With respect to the DSC measurements, experimental details were added to the Methods section in the Supplementary Information.

PCE11 and PCBM were dissolved in CB:DCB (1:1) at a concentration of 10 mg/mL and stirred in a glovebox at 80°C for overnight. The solutions were mixed with different PCE11:PCBM ratios and again stirred at 80°C for overnight. The homogeneous solutions were dried on clean substrates in a glovebox at 80°C. For each measurement, 3-5 mg material was collected from the substrate and filled into a DSC pan. DSC measurements were carried out using a Q1000 DSC setup from TA Instruments. The temperature of conventional DSC measurements ranges from -50°C to 310°C with a heating and cooling rate of 10 K min⁻¹. For the modulated DSC measurement, the temperature ranges from -50°C to 310°C with a heating and cooling rate of 3 K min⁻¹ and a modulated temperature of ±1 K min⁻¹.

It has to be underlined that PCE11 strongly aggregates in solution at temperatures lower than 70°C. Different from the DSC sample preparation report previously,⁴ the PCE11:PCBM solutions were dried at 80°C. As the drying kinetics of the PCE11:PCBM DSC samples is strongly related to the materials processing, PCE11:PCBM composites might deliver different miscibility information from their 1st DSC heating scans. To avoid any influence during sample preparation and to analyse the interaction between PCE11 and PCBM, the 2nd DSC heating scans of

PCE11:PCBM blends were used for analysing their thermal behaviour. Nevertheless, the complete DSC scans of PCE11:PCBM 1:1 blend were added to the manuscript. As shown in **Figure R10**, two distinct melting peaks were found from the 1st heating scan of the PCE11:PCBM 1:1. The melting peak at $\sim 281^{\circ}\text{C}$ was contributed by the crystallites of PCE11 and PCBM, while the one at $\sim 257^{\circ}\text{C}$ was contributed by the imperfect crystallites of PCE11:PCBM blends. Although both melting peaks were detected from the 2nd heating scan of PCE11:PCBM 1:1 blend, the nature of crystallites was changed upon the melting and cooling processes. With respect to the 2nd heating scan of the PCE11:PCBM 1:1 blend, the melting peak at $\sim 285^{\circ}\text{C}$ was solely composed by the crystallites of PCBM, while the one at $\sim 254^{\circ}\text{C}$ was mainly composed by the imperfect crystallites of PCE11 mixed with PCBM. The melting and cooling enthalpy change of PCE11:PCBM 1:1 blend and the corresponding neat materials are summarized in the **Table R3**.

We fully agree with the reviewer that the enthalpy change of PCE11:PCBM blends would give valuable information on the molecular order. However, it might be a bit difficult to explore much information from the PCE11:PCBM blends, as the melting peak of PCE11 is strongly overlapped with that of PCBM at $\sim 281^{\circ}\text{C}$. As mentioned above, the melting peak at $\sim 281^{\circ}\text{C}$ (1. heating scan) represents the melting process of both PCE11 and PCBM crystallites. We could distinguish from the 2nd heating scan that the melting peak at $\sim 285^{\circ}\text{C}$ consisted of PCBM crystallites, and the one at $\sim 254^{\circ}\text{C}$ mainly of the PCE11:PCBM mixture crystallites. As the melting temperature of PCE11 crystallites is overlapped with that of PCBM crystallites, the melting peak at 281°C for PCE11:PCBM 1:1 blend cannot be deconvoluted to extract more information. Although detailed analysis of the thermal behaviour of the PCE11:PCBM blends from DSC measurements is already out of the scope of this manuscript, the aforementioned discussion along with Figure R10 and Table R3 was added to the Supplementary Information.

Figure R10 thermal behavior of PCE11, PCBM and PCE11:PCBM 1:1 blend measured from DSC heating and cooling scans. Scale bar: 0.5 W/g.

Table R3 Enthalpy change of PCE11, PCBM and PCE11:PCBM 1:1 blend. ⁽¹⁾ Melting peak at low temperature. ⁽²⁾ Melting peak at high temperature.

	PCE11	PCBM		PCE11:PCBM	
		(1)	(2)	(1)	(2)
1. Heating	30.24 J/g	7.298 J/g	18.05 J/g	11.85 J/g	7.016 J/g
2. Heating	24.08 J/g	2.427 J/g	10.14 J/g	10.90 J/g	2.984 J/g
1. Cooling	25.56 J/g	9.605 J/g		17.13 J/g	

In addition, with the neat polymer featuring such a small supercooling and well defined crystallisation peak in the 1st cooling in the neat polymer, I would suggest to analyse what occurs with this feature upon blending and casting for all the blends. Does the supercooling increase (which means ordering starts to be hindered). Is its enthalpy getting reduced, etc.?

- As suggested by the reviewer, we plotted the first cooling scans of PCE11:PCBM blends and neat materials in **Figure R11**, and summarized the enthalpy changes extracted from the crystallization peaks of PCE11:PCBM blends and neat materials in

Table R4. The ΔH of neat PCE11 and neat PCBM were determined to be 25.56 J/g and 9.605 J/g, respectively. As discussed in-depth in the next section, the melting temperature of the PCE11 crystallites was easily influenced by addition of small amounts of PCBM (10-15 %), and the eutectic point at $\sim 253^\circ\text{C}$ was found at very high PCE11 loading (85-90 %). With respect to the DSC cooling scans of PCE11:PCBM blends, the crystallization peaks of PCE11 crystallites (or more exactly, the PCE11:PCBM mixed crystallites) were found at $\sim 234^\circ\text{C}$ with ΔH of ~ 20 J/g. With increasing PCBM loading from 0 to 50%, ΔH of crystallization reduced from 25.56 J/g for neat PCE11 to 17.13 J/g for a PCE11:PCBM 1:1 blend.

We fully agree with the reviewer that the “super-cooling” of PCE11 crystallites do influence the formation of PCE11:PCBM mixed crystallite in the blends. As shown in **Figure R11**, all the PCE11:PCBM blends exhibited similar well defined crystallization peaks with the contribution mainly from PCE11. Although the reduction in ΔH for PCE11:PCBM blends can be attributed to impurities in PCE11 crystallites rising from PCBM addition, the formed PCE11:PCBM mixed crystallites exhibited comparable high ΔH when correction for the volume fraction of PCBM. Moreover, it’s very important to point out that from the sample with 30% PCE11 loading the contribution from both PCBM and PCE11 crystallites can be well resolved. The ΔH of PCE11:PCBM with 30% PCE11 loading was determined to be 21.95 J/g, which is significantly higher than that of PCBM (9.605 J/g), as well as the calculated value by simply taking into account the volume fraction of PCE11 and PCBM crystallites (14.39 J/g). To summarize, the “super-cooling” effect of PCE11 facilitates the formation of the mixed PCE11:PCBM crystallites, especially for the blends with high PCBM loadings.

Figure R11 The 1st cooling scans of PCE11:PCBM blends.

Table R4 Enthalpy change ΔH of PCE11:PCBM blends and neat materials extracted from the 1st cooling scan.

	ΔH
PCE11	25.56 J/g
70% PCE11	19.74 J/g
50% PCE11	17.13 J/g
30% PCE11	21.95 J/g
PCBM	9.605 J/g

3) The above is important as I think the authors refer to spinodal donor-acceptor demixing as they like to suggest that demixing of the two materials occurs only in the amorphous phase. This is not that clearly spelled out in the manuscript – if the authors want to stick to this hypothesis. In my opinion, it may be, but it may not; based on the data provided I am not convinced. It still could be order-induced phase separation – just molecular ordering on smaller length scales than e.g. accessible with X-ray diffraction. I think it will be difficult to obtain data that solves this issue fully. I am not sure if it is needed at this stage. The burn-in is clear and I fully agree that it originates from phase separation. I personally would rephrase this part and may take out the word ‘spinodal’: i.e. I would suggest to simply go for ‘Abnormal

strong burn-in degradation of highly efficient polymer solar cells caused by donor-acceptor blends.' I know some calculations are presented supporting the spinodal idea, however, I wonder how much the graph presented in Figure 4 changes when slightly different parameters are used?

- We greatly thank the reviewer for the important comments, and fully agree that it was not easy to identify the type of phase separation based on the data shown in the initial manuscript. On the one hand we want to point out the importance of the EL investigation. In addition, we performed GISAXS measurements to analyze the morphology change in the amorphous region of PCE11:PCBM blends. Based on the experimental data collected from GISAXS, GIWAXS, EL and FTPS measurements, we could finally conclude that the drastic reduction in J_{SC} of PCE11:PCBM solar cells is indeed induced by the donor-acceptor phases demixing in the solid film. This demixing occurs even at room temperature in the dark, and is only related to the finely-mixed amorphous regions.

Moreover, to clearly present and highlight the significance of the work on theoretical calculation, the work flow of the calculation process is schematic illustrated in **Figure R12**. Starting from the Molecular Structure, the required information can be step by step calculated for predicting the miscibility of two components. The calculated molecular parameters are in great accordance with the experimental values as well as the values reported in literature. It is worthwhile to again underline that this work not only delivers the information on the demixing of donor and acceptor phases in PCE11:PCBM system, but also demonstrates an elaborated protocol on characterizing, analyzing and predicting the phase behavior and the miscibility of donor and acceptor materials. This protocol, which is a very powerful tool for design and development of next generation OPV systems with promising stability and reliability, is definitely of great significance and interest to the community.

Figure R12 Workflow of the theoretical calculation used for predicting the phase behavior and miscibility of donor and acceptor systems.

Some additional comments:

1) The thermal analysis data presented in Fig S7b indicates that PCE11:PCBM features a eutectic phase diagramme as the P3HT:PCBM binary does, with a eutectic point between 85 and 90 % PCE11 (so, at a much higher donor content than for P3HT:PCBM system). This phase behaviour implies that both melting points are depressed; the one of PCE11 seems more strongly depressed as the eutectic point is so much shifted towards the polymer rich side. Eutectic temperatures is around 250 C, hence the shoulder on the higher-temperature side for the 85% blend indicates that this feature is the PCBM melting – i.e. in my opinion also the PCBM melting is depressed. Clearly, having some more data for the compositions between 15% to 85 % will assist to identify correctly the eutectic composition. This will have significance whether it is or not spinodal decomposition.

- We greatly thank the reviewer for the helpful comments. The required DSC heating scans of PCE11:PCBM blends are summarized in **Figure R13a**. According to the DSC data, we fully agree with the reviewer that the melting point depression was observed for both PCE11 and PCBM crystallites. The crystallites of PCE11 are more strongly depressed than that of PCBM. The eutectic point was found at ~253°C, and

can be already resolved from the blends with 85-90% PCE11, indicating that the thermal behavior of PCE11 crystallites were easily influenced by adding small amount of PCBM. The melting peak of PCBM crystallites can still be detected for the blends with up to 60% PCE11. This is significantly different from the P3HT:PCBM system previously reported by us.⁴ As shown in **Figure R13b**, the melting peak of PCBM crystallites can only be detected for the blends with 0-30% P3HT. The difference in thermal behavior of PCE11:PCBM and P3HT:PCBM reveals that P3HT is more miscible with PCBM than PCE11, which is in excellent agreement with the findings demonstrated in this work.

Figure R13 (a) the 2nd DSC heating scans of PCE11:PCBM blends; (b) the 2nd DSC heating scans of P3HT:PCBM blends.⁴

- 2) I do not want to suggest even more experiments, but out of curiosity, have the authors performed measurements with ICBA with a higher T_g and it seems somewhat better miscibility with P3HT?
 - We fully agree on this useful suggestion. We tried to fabricate and optimize the solar cells based on PCE11:ICBA. However, the PCE11:ICBA did not exhibit satisfied performance compared to the PCE11:PCBM control cells, as shown in **Figure R14**. Although reasonable high V_{OC} was obtained for the PCE11:ICBA due to the preferable energetic levels, the significantly low J_{SC} and FF have to be attributed to the insufficient charge carrier dissociation and elevated charge recombination, which is strongly related to the morphological properties of the PCE11:ICBA blend.

Figure R14 (a) J - V characteristics of PCE11:PCBM and PCE11:ICBA; (b) statistic photovoltaic parameters collected from 12 PCE11:ICBA solar cells.

Due to the very poor performance obtained from the PCE11:ICBA solar cells, it's not reasonable to experimentally verify the device stability. Nevertheless, we used the approach demonstrated in this work to theoretically predict the miscibility of ICBA and polymer donors. The molecular parameters of P3HT, PCE11, PCBM and ICBA used for calculation are listed in the **Table R5**.

Table R5 The calculated molecular weight, molar volume, liquid density and Hildebrand parameter δ_T for P3HT, PCE11, PCBM and ICBA. * Experimental values estimated in our lab or in literature.⁵⁻⁷

	Molecular Weight (g mol ⁻¹)	Molecular Volume (cm ³ mol ⁻¹)	Liquid Density (g cm ⁻³)	δ_T (MPa ^{1/2})
P3HT	173.80/166.1*	148.668/151.0*	1.17/1.1*	19.36
PCE11	226.354	187.168	1.21	19.14*
PCBM	910.89/910.5*	548.041/607.0*	1.66/1.5*	21.60
ICBA	936.84/952.5*	542.652/635.0*	1.72/1.5*	20.81

The interaction parameters summarized in **Table R6** were calculated for P3HT:PCBM, P3HT:ICBA, PCE11:PCBM and PCE11:ICBA according to the following Equation from regular solution theory⁸:

$$\chi_{1,2} = \frac{v_0}{RT} (\delta_{T1} - \delta_{T2})^2$$

where $\chi_{1,2}$ is the polymer-fullerene interaction parameter, v_0 is the molar volume of the lattice site in the Flory-Huggins model. The entropic contribution is usually between 10^{-6} and 10^{-2} , which is smaller in magnitude than the enthalpic contribution

given for polymer solvent (approximately 0.34).⁹⁻¹¹ Therefore, the entropic contribution was not taken into account for calculation.

Table R6 Interaction parameter $\chi_{1,2}$ estimated for P3HT:PCBM, P3HT:ICBA, PCE11:PCBM and PCE11:ICBA.

	Calculated $\chi_{1,2}/v_0$
P3HT:PCBM	0.00202
P3HT:ICBA	0.00085
PCE11:PCBM	0.00244
PCE11:ICBA	0.00112

spinodal interaction parameter $\chi_{spinodal}$ that defines the boundary between the two-phase region and homogenous region can be derived⁷:

$$\chi_{spinodal} = \frac{v_0}{2} \left(\frac{\rho_1}{M_1\phi_1} + \frac{\rho_2}{M_2(1-\phi_1)} \right)$$

where v_0 is the molar volume of the lattice site in the Flory-Huggins model; ρ_1 , M_1 and ϕ_1 are the density, the molecular weight and the fraction volume of fullerene acceptor, respectively; ρ_2 and M_2 are the density and the molecular weight of polymer, respectively. As the PCBM and ICBA have similar molecular weight and density, the values of $\chi_{spinodal}/v_0$ vs. fraction volume are very similar for P3HT:PCBM and P3HT:ICBA, as well as for PCE11:PCBM and PCE11:ICBA. However, as depicted in **Figure R15**, the interaction parameter $\chi_{1,2}/v_0$ of P3HT:PCBM is more close to the spinodal curve $\chi_{spinodal}/v_0$ than that of P3HT:ICBA, indicating that the P3HT:PCBM is expected to be less miscible than P3HT:ICBA. The same to the PCE11-based systems, PCE11:PCBM is expected to be more miscible than PCE11:ICBA as well. A recent publication by Zhan et al. confirmed that the thermal stability of PCE11:PCBM can be significantly improved by adding small amounts of ICBA into the active blends,¹² which is in excellent accordance with the findings demonstrate in this section.

Figure R15 The polymer/fullerene liquid (melt) solid transition diagrams estimated for P3HT:PCBM, P3HT:ICBA, PCE11:PCBM and PCE11:ICBA as a function of the volume fraction of polymer. The dashed lines that indicate the interaction parameters of polymer-fullerene blends are taken from Table R6.

3) Finally, I think the authors should include some important references: on mixing, initial work from the Ade/McNeill groups, Treat/Chabinye et al., Westacott/Stingelin, Russell group at UMASS, the group of Rasmus Schroeder in Germany; on spinodal decomposition of OPV blends: Steiner/Friend; phase diagrams: Nelson/Stingelin, used also by Hadziiaonnu. UV-vis: Work by Spano et al.

- We thank the reviewer for pointing out the pioneer works in literature. The mentioned publications were added to the manuscript, and highlighted in yellow.

In summary, I think the present work is intriguing and a broad audience will be interested in it and will learn from it. Hence, I strongly suggest publication – after issues pointed to above have been clarified or addressed.

Referee: 3**Comments to the Author**

The author demonstrated strong burn-in degradation in highly-efficient state-of-the-art OSCs induced by spinodal demixing of the donor and acceptor phases, which dramatically reduces charge generation and can be attributed to the inherently low miscibility of both materials. In my opinion, the topic of "stability of OSCs" is very important but the novelty of this paper is not enough for Nature Communications. I think Scientific Reports is a good choice for this paper. Some comments are shown as below:

1. We already know the reason of this degradation (metastable state of morphology), but we don't know how to overcome it. That's the point!
 - We'd like to thank the reviewer for time and effort spent in evaluating our work. However, we politely disagree with the reviewer's opinion that this manuscript lacks in novelty or impact for publication in Nature Communications. As follows, we summarize in short why we believe that this work is indeed of great significance and interests to the community and the broad readership of Nature Communications.

We demonstrate in this work that the abnormal strong burn-in loss of PCE11:PCBM solar cells, which occurred even at room temperature in the dark, can be ascribed to a spontaneous spinodal demixing of the amorphous donor and acceptor phases. Systematic investigations have been carried out to directly and indirectly verify the microstructure changes in the PCE11:PCBM BHJ morphology. Based on the elaborated experimental data and analysis summarized in the manuscript, we can finally concluded that donor-acceptor demixing in the amorphous regions is indeed caused by the inherent low miscibility of the two components, and is therefore inevitable for such highly efficient OPV system. This information is, for the first time, clearly demonstrated for the current generation of state-of-the-art OPV systems and is of great importance to the research community.

Moreover, as the abnormal burn-in loss is induced by the inherently low miscibility of the donor and acceptor materials, design and development of novel organic materials with promising efficiency and proper miscibility will be the most elegant way to solve these microstructure meta-stability issues. In this work, we demonstrate for the first time a systematic procedure allowing to predict the stability of a BHJ microstructure based on a combined experimental / theoretical approach. This procedure is not only a

powerful tool for designing and developing the next generation OPV material systems with superior stability and reliability, but also very useful to understand the phase-behaviour, limits and potential of all kinds of organic electronic devices, such as organic photodetectors, organic light emitting diodes, etc.

2. There is only one high-efficiency polymer/fullerene pair discussed in this paper, it's not enough to make a strong conclusion based on one case.

- We thank the reviewer for the comment, but politely disagree with the reviewer's opinion that the conclusion of our work is not strong enough. This work demonstrates on the abnormal strong burn-in loss observed for PCE11:PCBM system. The drastic reduction in J_{SC} of highly efficient solar cells, which occurred even at room temperature in the dark, is ascribed to the demixing of the amorphous donor and acceptor phases. As this conclusion was made based on systematic investigations, including temperature-dependent $J-V$ characterization, EL, FTPS, GIWAXS and GISAXS data, we believe that the messages and conclusion delivered by this work are very strong and rigid.

We don't think the observed abnormal strong burn-in loss needs to be proven using other high-efficiency polymer/fullerene pairs, as this is not the general case for all OPV material systems. As also accepted by the reviewer, the degradation is usually induced by the metastable state of BHJ morphology, which changes upon external stress such as thermal annealing or solvent vapor annealing. However, this work demonstrates an extreme case observed for the state-of-the-art OPV material system, in which the finely-mixed donor-acceptor amorphous regions phase-separated even at room temperature in the dark, owing to the poor inherent miscibility of the two components. This information, which is not only related to OPV technology, but also very useful to understand the limits and potential of all kinds of organic electronic devices, must be delivered to the community, especially to chemists and materials scientists for future design of highly efficient and stable organic materials.

3. After the degradation of OSCs, the V_{OC} always higher. Please add some discussion.

- It is generally accepted that the V_{OC} of OPV devices is directly related to the effective bandgap E_{eff} of BHJ system, which is defined by the following equation¹³:

$$E_{eff} = E_{CT} - 2w$$

where E_{CT} is the band maximum of the charge transfer absorbance, and w is the width of the Gaussian fitting for the CT absorbance.

According to Vandewal et al., an excellent correlation was found to define the V_{OC} of OPV systems, as shown in the following equation¹³:

$$V_{oc} \approx \frac{E_{eff}}{e} - 0.43 \text{ V}$$

The effective bandgap E_{eff} is strongly related to the nature of the donor-acceptor interfaces in BHJ blends. By reducing the donor-acceptor interface area, the V_{OC} of solar cells can be correspondingly enhanced.¹⁴ Coming back to the findings demonstrated in this work, we cannot quantitatively determine the change in E_c for the aged PCE:PCBM samples, because (1) the FTPS characteristics of the aged PCE11:PCBM didn't feature a clear CT band for Gaussian fitting; (2) the EL spectra of PCE11:PCBM BHJ samples are strongly overlapped with that of neat PCE11, which causes problem in precisely determining the band width and band maximum of the CT state for aged PCE11:PCBM BHJ sample. As the correlation between the E_{eff} and V_{OC} of solar cells is well known to the OPV community, the discussion on V_{OC} changes for aged PCE11:PCBM samples is beyond the scope of the current work, we didn't add the above discussion to the previous manuscript.

To summarize, we greatly acknowledge the time and effort the 3rd reviewer spent in evaluating our current work. However, we politely disagree with the reviewer's opinion that our work is not relevant enough for publication in Nature Communications. We hope that our response to all reviewers could also convince reviewer 3 that spinodal demixing in the amorphous phase of polymer-fullerenes blends is indeed a most crucial process determining the long time stability. The origin of the spinodal demixing lies in the low miscibility of the two components, and the direct consequence on the device performance is a fast demixing and J_{SC} burn-in. Theoretical calculations of the interaction parameters underline the importance of miscibility and suggests to further explore this as a design parameter for developing the next generation of stable and efficient solar cell materials. We hope that this may convince reviewer 3 that this work is of great significance and broad interest to the research community.

References:

- 1 Taylor, C. A. X-ray diffraction methods in polymer science by L. E. Alexander. *Journal of Applied Crystallography* **3**, 428-428, (1970).
- 2 Alexander, L. X-ray diffraction methods in polymer science. *J Mater Sci* **6**, 93-93, (1971).
- 3 Lin, T. L. *et al.* Effect of Arm Length on the Aggregation Structure of Fullerene-Based Star Ionomers. *The Journal of Physical Chemistry B* **108**, 14884-14888, (2004).
- 4 Li, N., Machui, F., Waller, D., Koppe, M. & Brabec, C. J. Determination of phase diagrams of binary and ternary organic semiconductor blends for organic photovoltaic devices. *Sol Energ Mat Sol C* **95**, 3465-3471, (2011).
- 5 Perea, J. D. *et al.* Combined Computational Approach Based on Density Functional Theory and Artificial Neural Networks for Predicting The Solubility Parameters of Fullerenes. *The Journal of Physical Chemistry B* **120**, 4431-4438, (2016).
- 6 Ulum, S. *et al.* The role of miscibility in polymer: fullerene nanoparticulate organic photovoltaic devices. *Nano Energy* **2**, 897-905, (2013).
- 7 Kozub, D. R. *et al.* Polymer crystallization of partially miscible polythiophene/fullerene mixtures controls morphology. *Macromolecules* **44**, 5722-5726, (2011).
- 8 Rubinstein, M. & Colby, R. H. *Polymer Physics*. (Oxford University Press, 2003).
- 9 Graessley, W. W. *et al.* Regular and Irregular Mixing in Blends of Saturated Hydrocarbon Polymers. *Macromolecules* **28**, 1260-1270, (1995).
- 10 Maranas, J. K. *et al.* Liquid Structure, Thermodynamics, and Mixing Behavior of Saturated Hydrocarbon Polymers. 1. Cohesive Energy Density and Internal Pressure. *Macromolecules* **31**, 6991-6997, (1998).
- 11 Müller, M. Miscibility behavior and single chain properties in polymer blends: a bond fluctuation model study. *Macromolecular Theory and Simulations* **8**, 343-374, (1999).
- 12 Cheng, P. *et al.* Alloy Acceptor: Superior Alternative to PCBM toward Efficient and Stable Organic Solar Cells. *Advanced Materials* **28**, 8021-8028, (2016).
- 13 Vandewal, K. *et al.* The Relation Between Open-Circuit Voltage and the Onset of Photocurrent Generation by Charge-Transfer Absorption in Polymer : Fullerene Bulk Heterojunction Solar Cells. *Advanced Functional Materials* **18**, 2064-2070, (2008).
- 14 Vandewal, K. *et al.* Increased open-circuit voltage of organic solar cells by reduced donor-acceptor interface area. *Adv Mater* **26**, 3839-3843, (2014).

Reviewers' comments:

Reviewer #1 (Remarks to the Author):

Decision: Accept pending minor revisions.

I am broadly satisfied with the revisions to the manuscript. It now makes a much stronger case for the origin of the burn-in behaviour and its implications. A small number of revisions are requested before the manuscript is suitable for publication in Nature Communications.

- 1) Solar cell performance metrics for the fresh device batches should be tabulated in the supporting information (S.I.).
- 2) References to '1 sun illumination' should read '1 sun equivalent illumination' because the LED output does not match the AM 1.5 spectrum.
- 3) Lines 166-169. The T_g value of 14.6°C should be treated with more caution than the authors currently communicate because the m-DSC measurements do not replicate the thermal histories of the blend films used for the solar cell devices. The absence of DIO in the m-DSC samples should also be explicitly noted. Please address this in the main text.
- 4) The EL spectra corresponding to the fresh and aged BHJ samples should also be presented at constant current density (e.g. as a complementary figure in the S.I.). Does the aged BHJ sample have higher electroluminescence external quantum efficiency?

Reviewer #2 (Remarks to the Author):

The authors have made a thorough revision of the manuscript and have included a wealth of new data. I thus suggest to accept this manuscript for publication after on final comment is amended:

On page 12 the authors mention 'a supercooling effect'. Supercooling is not an effect and it does not dictate what phase forms. It simply states how large the difference is between the melting (or dissolution) and the crystallization. If this difference is small (i.e. crystallization occurs at very similar temperatures as the melting) this implies a small supercooling and that crystallization is not hindered. Neat PCE 11 crystallizes easily as evident from the small supercooling (i.e. small difference between melting and crystallization). This is different with the blends. In short, the authors should amend that sentence on supercooling as the way it currently stands, it does not make sense - indeed, is just blatantly wrong; which would be a shame for the manuscript..

A point-by-point Response to Reviewers' Comments:

Reviewers' comments:

Reviewer #1 (Remarks to the Author):

Decision: Accept pending minor revisions.

I am broadly satisfied with the revisions to the manuscript. It now makes a much stronger case for the origin of the burn-in behaviour and its implications. A small number of revisions are requested before the manuscript is suitable for publication in Nature Communications.

- We greatly thank the reviewer for the very helpful comments.

1) Solar cell performance metrics for the fresh device batches should be tabulated in the supporting information (S.I.).

- The following information was added to the supplementary information, Table S1.

Table S1 photovoltaic parameters of fresh and aged PCE11:PCBM solar cells.

	V_{oc} [V]	J_{sc} [mA cm ⁻²]	FF [%]	PCE [%]
Fresh	0.74	17.84	69.7	9.20
Aged	0.73	11.78	65.4	5.62

2) References to '1 sun illumination' should read '1 sun equivalent illumination' because the LED output does not match the AM 1.5 spectrum.

- The description was revised according to the reviewer's suggestion.

3) Lines 166-169. The T_g value of 14.6°C should be treated with more caution than the authors currently communicate because the m-DSC measurements do not replicate the thermal histories of the blend films used for the solar cell devices. The absence of DIO in the m-DSC samples should also be explicitly noted. Please address this in the main text.

- The description in page 8 was modified as follows:

“We attempted to determine the glass transition temperature (T_g) of neat materials and BHJ blends by means of temperature-modulated differential scanning calorimetry (m-DSC). However, as depicted in Supplementary Fig. 6, no clear T_g was observed for neat PCE11 and neat PCBM, while a reversible transition at 14.6 °C was observed for the PCE11:PCBM 1:1 blend from the 2nd heating scan. It’s worthwhile to mention that the m-DSC measurement cannot mimic the thermal behaviour of PCE11:PCBM BHJ blend used for OSCs, and the m-DSC samples prepared from drop-casting without solvent additive may have different morphology compared to that optimized for OSCs.”

4) The EL spectra corresponding to the fresh and aged BHJ samples should also be presented at constant current density (e.g. as a complementary figure in the S.I.). Does the aged BHJ sample have higher electroluminescence external quantum efficiency?

- The EL spectra of fresh and aged PCE11:PCBM samples measured at an external constant current of 50 mA were added to the S.I., Figure S7. The aged BHJ sample shows higher EL external quantum efficiency than the fresh sample, as the singlet emission of PCE11 and PCBM are also involved in the emission spectrum of the aged BHJ sample.

Reviewer #2 (Remarks to the Author):

The authors have made a thorough revision of the manuscript and have included a wealth of new data. I thus suggest to accept this manuscript for publication after on final comment is amended:

On page 12 the authors mention 'a supercooling effect'. Supercooling is not an effect and it does not dictate what phase forms. It simply states how large the difference is between the melting (or dissolution) and the crystallization. If this difference is small (i.e. crystallization occurs at very similar temperatures as the melting) this implies a small supercooling and that crystallization is not hindered. Neat PCE 11 crystallizes easily as evident from the small supercooling (i.e small difference between melting and crystallization). This is different with the blends. In short, the authors should amend that sentence on supercooling as the way it currently stands, it does not make sense - indeed, is just blatantly wrong; which would be a shame for the manuscript.

- We greatly thank the review for the very helpful comments as well as the time and effort in evaluating our work. The description was corrected in the manuscript.

“To summarize, the well-defined crystallization of PCE11 facilitates the formation of the mixed PCE11:PCBM crystallites, especially for the blends with high PCBM loadings.”